# AutoPR: Let's Automate Your Academic Promotion!

## Abstract

As the volume of peer-reviewed research surges, scholars increasingly rely on social platforms for discovery, while authors invest considerable effort in promoting their work to ensure visibility and citations. To streamline this process and reduce the reliance on human effort, we introduce Automatic Promotion (AutoPR), a novel task that transforms research papers into accurate, engaging, and timely public content. To enable rigorous evaluation, we release PRBench, a multimodal benchmark that links 512 peer-reviewed articles to high-quality promotional posts, assessing systems along three axes: Fidelity (accuracy and tone), Engagement (audience targeting and appeal), and Alignment (timing and channel optimization). We also introduce PRAgent, a multi-agent framework that automates AutoPR in three stages: content extraction with multimodal preparation, collaborative synthesis for polished outputs, and platform-specific adaptation to optimize norms, tone, and tagging for maximum reach. When compared to direct LLM pipelines on PRBench, PRAgent demonstrates substantial improvements, including a 604% increase in total watch time, a 438% rise in likes, and at least a 2.9x boost in overall engagement. Ablation studies show that platform modeling and targeted promotion contribute the most to these gains. Our results position AutoPR as a tractable, measurable research problem and provide a roadmap for scalable, impactful automated scholarly communication.

## 1 Introduction

Large-scale pretrained AI models have recently advanced automated reasoning in academic settings, fueling AI4Research applications and a marked rise in scholarly assistant (Chen et al., 2025a;b; Eger et al., 2025; Zhou et al., 2025). Therefore, as shown in Figure 1 (a), the number of accepted conference papers has increased sharply (White, 2019; Azad & Banu, 2024). With this surge, researchers cannot feasibly track all relevant papers across conferences (Murayama et al., 2016; Ferguson & Fenner, 2021). To obtain information more efficiently, readers increasingly rely on social media and digital platforms to keep up with current developments (Kulczycki, 2013; Jucan & Jucan, 2014; Davies & Hara, 2017). Meanwhile, authors proactively promote their work to expand visibility, attract citations, and increase influence (Collins et al., 2016; Gudi & Basker, 2019; Mulimani, 2024). However, as shown in Figure 1 (b), without promotion (PR), both influence and citations decline (Betz et al., 2023; Weissburg et al., 2024), yet producing high-quality promotion materials still depends on manual effort and substantial time and cost (see Figure 1 (c)) (Earlham Institute, 2023; Maron et al., 2016).

Recently, intelligent agent systems, which make autonomous decisions and adapt actions, have shown promise in academic contexts (Hu et al., 2024; Gridach et al., 2025; Wei et al., 2025). By automating research-promotion tasks such as generating concise summaries, designing visual abstracts, and conducting targeted promotion, these agents can increase the visibility and impact of scholarly work while reducing human effort (Lu et al., 2024; Sun et al., 2025; Zhang et al., 2025). However, a systematic benchmark for automated academic promotion on social platforms is still lacking. Current research offers neither a comprehensive evaluation of LLMs on end-to-end promotion tasks nor complete pipelines for transforming academic papers into effective multimodal promotion materials.

To fill this research gap, as illustrated in Figure 1 (d), we first introduce a **novel task, AutoPR**, which automatically generates academic promotion content. To support evaluation, we construct the **Academic Promotion Benchmark (PRBench)**, which links 512 peer-reviewed articles across

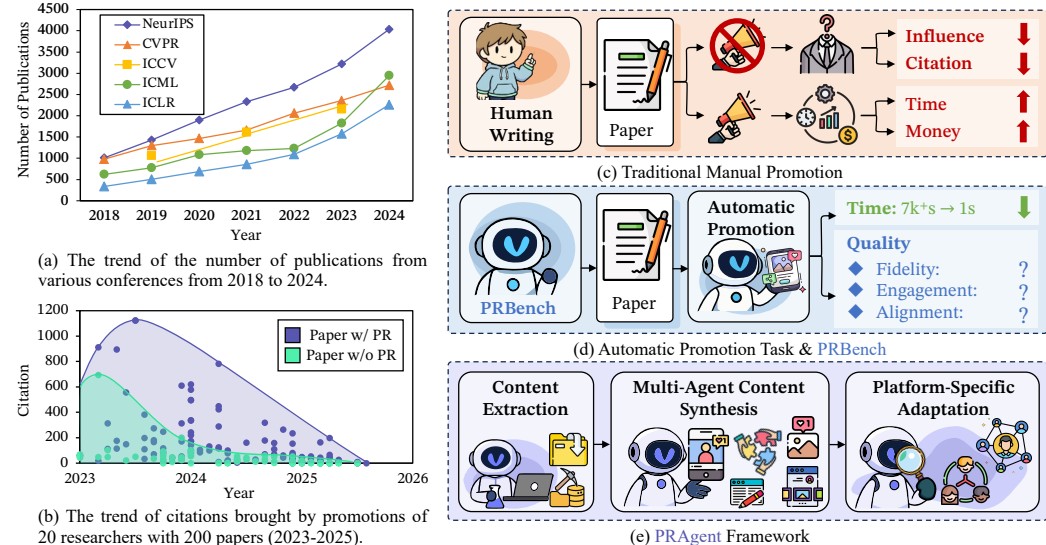

(a) The trend of the number of publications from various conferences from 2018 to 2024.

(b) The trend of citations brought by promotions of 20 researchers with 200 papers (2023-2025).

(c) Traditional Manual Promotion

(d) Automatic Promotion Task & PRBench

(e) PRAgent Framework

Figure 1: Overview of our study: Automatic Promotion (AutoPR) task, its benchmark PRBench, and the associated method PRAgent. The details of citation trend analysis are shown in Appendix C.

disciplines with curated multimodal promotion materials. We systematically assess agent performance along three dimensions: (i) Fidelity: producing accurate, persuasive content with proper tone and length; (ii) Engagement: identifying and involving stakeholders such as academic peers, journalists, and policymakers; and (iii) Alignment: timing dissemination based on audience behavior and channel dynamics. Our analysis of current agent frameworks reveals persistent limitations in contextual understanding and targeting precision for these tasks.

To overcome these challenges and provide an end-to-end pipeline, as shown in Figure 1 (e), we further present **PRAgent**, a three-stage framework for scholarly promotion: (1) *Content Extraction* applies hierarchical summarization and multimodal processing to create concise paper summaries, social media posts, and graphical abstracts. (2) *Multi-Agent Content Synthesis* uses a collaborative agent system to refine extracted information into polished outputs, transforming structured materials into coherent promotion-ready content. (3) *Platform-Specific Adaptation* models platform-specific preferences, allowing PRAgent to adjust tone and tagging to maximize user engagement. We evaluate PRAgent on the PRBench against standard LLM pipelines, showing much optimized content accuracy, engagement, and platform alignment. In real-world application, it shows a 604% increase in total watch time and a 438% increase in likes on real social media. These findings demonstrate PRAgent's effectiveness and chart a path toward automated scholarly communication.

Our contributions can be summarized as follows:

- **Novel AutoPR Task:** We first formalize automatic academic PR (AutoPR) as a distinct research task with systematic evaluation metrics. We scope it as translating peer-reviewed research into tailored promotional materials, specifying inputs (manuscripts, figures, key findings) and outputs (press releases, social media posts, visual abstracts).

- **PRBench Dataset:** We present PRBench, a publicly released dataset of 512 paired multimodal samples linking peer-reviewed papers to their manually created PR posts across three AI-related fields, enabling rigorous end-to-end study of scholarly promotion.

- **PRAgent Framework:** We introduce PRAgent, a three-stage framework integrating Content Extraction, Multi-Agent Content Synthesis, and Platform-Specific Adaptation. Experiments on PRBench show PRAgent outperforms traditional LLM pipelines aross almost all LLMs. In real-world tests, it yields up to a 6x increase in total watch time, a 4x increase in likes.

## 2 TASK: AUTOPR

Here, we provide formal definition for Automatic Promotion (AutoPR) task. As shown in Figure 2, the objective is to automatically generate promotional content from a research doc-

Figure 2: The definition and overview of Automatic Promotion (AutoPR) Task.

ument, optimized for a specific audience and dissemination platform. Formally, a source research document $\mathbb{D} = (D_T, D_V, D_S)$ consists of the full text content $D_T$; a set of visual content $D_V = \{(v_1, c_1), (v_2, c_2), \dots, (v_n, c_n)\}$, where each pair $(v_i, c_i)$ consists of a visual (e.g., figure, table) and its corresponding caption; any supplementary materials $D_S$.

The dissemination target consists of two components: $\mathbb{T}_P$ is the target dissemination platform (e.g., Twitter, RedNote) and $\mathbb{T}_A$ is the intended audience (e.g., academic peers, general public). The task is to generate a promotional post $P$, which is a composition of text and visual elements tailored to the dissemination target. The generation process can be modeled as:

$$\hat{P} = \underset{P}{\arg\max} \, \mathbf{Pr}(P \mid \mathbb{D}, \mathbb{T}_P, \mathbb{T}_A). \tag{1}$$

The goal of this task is to find an optimal post $\hat{P}$ by simultaneously maximizing multiple objectives. This is a multi-objective optimization problem, as the core objectives are often in tension with one another. We define the objective function $\vec{F}(P)$ as:

$$\max_{\hat{P}} \vec{F}(P) = \max_{\hat{P}} \left\{ \alpha_1 \mathcal{S}_{\text{Fidelity}}(\hat{P} \mid \mathbb{D}) + \alpha_2 \mathcal{S}_{\text{Align}}(\hat{P} \mid \mathbb{T}_P) + \alpha_3 \mathcal{S}_{\text{Engage}}(\hat{P} \mid \mathbb{T}_A) \right\} \tag{2}$$

where the $\mathcal{S}_{\text{Fidelity}}(P \mid \mathbb{D})$ measures the factual accuracy and completeness of the post $P$ with respect to the source research document $\mathbb{D}$; $\mathcal{S}_{\text{Align}}(P \mid \mathbb{T}_P)$ evaluates how well the style, tone, and format of the post $P$ align with the norms and best practices of the target platform $\mathbb{T}_P$; $\mathcal{S}_{\text{Engage}}(P \mid \mathbb{T}_A)$ assesses the potential engagement of the post $P$ to capture the attention of and resonate with the target audience $\mathbb{T}_A$. $\alpha_i$ is a non-negative weight that controls the trade-offs between these objectives.

# 3 BENCHMARK: PRBENCH

This section introduces the Academic Promotion Benchmark (PRBench), a novel benchmark for evaluating intelligent agents on the task of automated academic promotion. In this section, we detail its construction, the evaluation protocol used, and the specific metrics derived from this protocol.

## 3.1 BENCHMARK CONSTRUCTION

The dataset was constructed through a three-stage process to ensure data quality, relevance, and utility for evaluating promotional agents. Detailed statistics are reported in Table 1 and Figure 3.

**Step 1: Data Collection** We first collected a corpus of papers from the arXiv repository submitted between June 2024 and June 2025, focusing on computer science subfields such as Computation & Language, Machine Learning, and Artificial Intelligence. In parallel, we retrieved related promotion posts for these articles from two major social media platforms: Twitter (X) and RedNote.

**Step 2: Data Pairing and Curation** To ensure all posts were human-authored, we first estimated their proportion of AI-generated content and excluded those with high AI likelihood. Next, we uniformly sampled 512 parallel pairs drawn from diverse sources and accounts. Each pair links a formal scientific artifact with its corresponding public-facing promotional material. The curation process required manual verification to ensure that each social media post directly promoted the associated arXiv paper. Each final pair includes both the research manuscript (PDF and metadata) and the promotional post (text and images).

| Statistic | Value |
|---|---|
| Avg. tokens per paper | 24,077.48 |
| Avg. figures/tables per paper | 20.91 |
| Avg. tokens per post | 543.30 |
| Avg. figures/tables per post | 5.37 |

Table 1: PRBench corpus statistics.

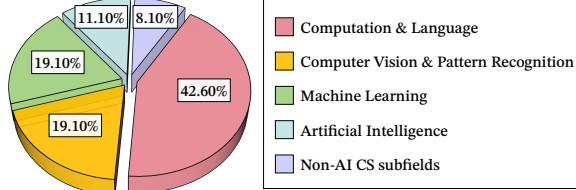

Figure 3: PRBench topic distribution.

**Step 3: Human Annotation and Quality Control**   To construct a reliable gold-standard ground truth, we implemented two expert-driven processes, with the full protocol detailed in Appendix D. (1) **Annotation for Fidelity Evaluation:** For each source paper, Gemini 2.5 Pro first generated a draft checklist of key factual points. A human expert then refined this checklist through corrections, additions, and deletions. Subsequently, three additional experts independently assigned importance weights from 1 (least critical) to 5 (most critical) to each fact. This procedure ensured both completeness and accurate representation of the paper's core contributions. (2) **Annotation for Engagement and Alignment Evaluation:** A panel of three experts independently annotated 512 authentic human-authored promotional posts. Each post was rated on a 0–5 scale according to the multi-dimensional criteria specified in Section 3.2. Small discrepancies ($\leq 1$) were resolved through averaging, while larger discrepancies were settled by consensus deliberation. The resulting scores provide the ground truth for comparing LLM and human assessments of content quality.

## 3.2 EVALUATION METRICS

To systematically evaluate the numerous subjective attributes of social media posts, we assess the intrinsic quality of the post itself using a scoring system, and evaluate external human interests via preference scores(see Appendix E for the specific evaluation prompts).

**LLM-as-Judge Calibration.**   For more realistic evaluation metrics, following Tan et al. (2024); Thakur et al. (2025), we calibrate human annotators and LLM evaluators with a shared 0–5 rubric across all Fidelity, Engagement and Alignment sub-metrics. The same rubric trains experts (Step 3 in Section 3.1) and conditions LLM prompts (Appendix E), ensuring that humans and models apply identical criteria. We select the judge model by correlating candidate LLM scores with these human ratings on 512 human-authored posts; Qwen-2.5-VL-72B-Ins shows the strongest and most consistent agreement and is used as the primary evaluator (see Section 5 and Appendix G). To reduce randomness and positional bias, we fix the LLM temperature at 0.01, a setting that gave the best human alignment, average three runs for each scalar score, and query each pairwise comparison twice with swapped order, treating disagreements as ties.

**Fidelity Evaluation.**   Inspired by Sun et al. (2025); Wu et al. (2025), the fidelity score is an average of two sub-metrics to measure factual accuracy and completeness: (1) *Authorship and Title Accuracy*, which assesses whether the post accurately and prominently presents the authorship and title. (2) *Factual Checklist Score*. For a post $P$ and source research document $\mathbb{D}$, we create a weighted factual checklist $\mathcal{C} = \{(c_1, w_1), \ldots, (c_n, w_n) \mid \mathbb{D}\}$. This checklist includes both fine-grained scientific claims and fundamental attribution facts. The *Factual Checklist Score* is calculated as:

$$\mathcal{S}_{\text{Checklist}}(P \mid \mathbb{D}) = \frac{\sum_{i=1}^{n} w_i \cdot v(P \mid c_i, \mathbb{D})}{\sum_{i=1}^{n} w_i}, \tag{3}$$

where $v(P \mid c_i, \mathbb{D})$ is the verdict from the LLM judge, a numerical score between 0 and 1.

**Alignment Evaluation.**   Informed by the theory of platform affordances which highlights the need for platform-specific strategies (Marabelli et al., 2018), alignment evaluation measures how well the generated content conforms to the norms and expectations of specific social media platforms $\mathbb{T}_P$. The sample's intrinsic alignment quality score is defined as the average rating across three criteria: (1) *Contextual Relevance* assesses the extent to which style, tone, and language align with platform norms and audience expectations. (2) *Visual–Text Integration* evaluates the effectiveness of coordination between textual and visual elements for the specific platform. These two metrics are constructed under the influence of P2P (Sun et al., 2025). (3) *Hashtag and Mention Strategy* examines the use of platform-specific hashtags and mentions to enhance reach and discoverability. The subjective preference alignment quality score is derived from the *Platform Interest*, in which the

Figure 4: overview of PRAgent, including: (1) Content extraction for preparing multimodal research material; (2) Multi-agent synthesis to transform structured data from Stage 1 into refined drafts; (3) Platform-specific adaptation to finalize the draft for publication.

post $P$ is evaluated against a reference post $P_{ref}$. This comparison simulates audience preferences to determine which post is more effective for platform-specific promotion and engagement.

**Engagement Evaluation.** Drawing from communication studies that define social media success through user engagement (Barger et al., 2016), this evaluation assesses the potential of the generated content to attract and interact with target audience $\mathbb{T}_A$. The sample's intrinsic engagement score is the average rating across four criteria: (1) ***Engagement Hook Strength*** evaluates the effectiveness of the opening in capturing attention and generating interest. (2) ***Logical Attractiveness*** assesses the clarity and coherence of the narrative in conveying the core message. (3) ***Visual Attractiveness*** scrutinizes the originality, aesthetic value, and informational contribution of visual elements. (4) ***Call-To-Action (CTA) Score*** measures the effectiveness of guiding the audience toward a desired next deeper action (e.g., reading the paper). The subjective preference engagement score is defined as the average win rate in pairwise comparisons under two perspectives: (1) ***Professional Interest*** evaluates the effectiveness in conveying scientific novelty and value to peers. (2) ***Broader Interest*** assesses clarity and appeal to a scientifically literate wider audience.

## 4 METHODOLOGY: PRAGENT

PRAgent is a multi-agent framework for the autonomous transformation of academic papers into platform-specific social media posts. As illustrated in Figure 4, the PRAgent workflow employs specialized agents across three stages: (1) Content Extraction and Structuring, (2) Multi-Agent Content Synthesis, and (3) Platform-Specific Adaptation and Orchestration. The detailed prompts for each agent are provided in Appendix F.

### 4.1 STAGE 1: CONTENT EXTRACTION

The initial stage converts unstructured PDF research documents ($D$) into structured, machine-readable formats via parallel textual and visual content pipelines.

#### 4.1.1 TEXTUAL CONTENT EXTRACTION AGENT

Due to frequent LLM context limitations, a structure-aware summarization strategy is applied by the Textual Content Extraction Agent: (1) **Structural Parsing**: The document $\mathbb{D}$ is first converted into intermediate HTML via `PyMuPDF`. Non-textual elements are then removed, and paragraph content is extracted, yielding the raw text $\mathbb{D}_T^{raw}$. (2) **Hierarchical Summarization**: It condenses the body text by adaptive hierarchical summarization. Content within the LLM's context window undergoes a single-pass summary. Longer texts are processed hierarchically by section: each chunk is independently summarized and recursively combined layer-by-layer. This method is formalized as:

$$\mathbb{D}_T^{sum} = \text{Summarize}(\text{Parse}(\mathbb{D}_T^{raw})), \tag{4}$$

where Summarize and Parse denotes the structural parsing and hierarchical summarization process described above, respectively.

### 4.1.2 VISUAL CONTENT PREPARATION AGENT

The Visual Content Preparation Agent manages the visual pipeline, identifying and pairing figures and tables with their captions. (1) **Image Conversion** (PDF2Img): First, we render each source PDF page into a high-resolution (250 DPI) PNG image. (2) **Layout Segmentation** (LayoutSeg): We utilize DocLayout-YOLO (Zhao et al., 2024) to perform layout analysis on each page image. This model detects bounding boxes for visual components (e.g., figure, table) and their captions. Detected components are subsequently cropped and saved. (3) **Component Pairing** (Pair): Then, we utilize a nearest-neighbor algorithm to associate visual elements with their captions and descriptions based on vertical proximity and a distance threshold. It yields a set of paired visual units, expressed as:

$$\mathbb{V}_{paired} = \text{Pair}(\text{LayoutSeg}(\text{PDF2Img}(\mathbb{D}))), \tag{5}$$

where $\mathbb{V}_{paired} = \{(v_1, c_1), (v_2, c_2), \ldots, (v_n, c_n)\}$, with $v_i$ being an extracted visual element and $c_i$ its corresponding caption and description.

## 4.2 STAGE 2: MULTI-AGENT CONTENT SYNTHESIS

The core of our framework is a collaborative multi-agent system that synthesizes and adapts content, transforming structured data from Stage 1 into polished drafts. This system comprises four distinct agents: Logical Draft Agent, Visual Analysis Agent, Textual Enriching Agent, and Visual-Text-Interleaved Combination Agent.

### 4.2.1 LOGICAL DRAFT AGENT

The Logical Draft Agent initiates content generation, converting summarized academic text ($\mathbb{D}_T^{sum}$) into a structured, factually accurate, and style-agnostic draft ($\mathbb{D}_T^{draft}$). Its operation is defined as:

$$\hat{\mathbb{D}}_T^{draft} = \mathcal{M}_{text}(D_T^{draft}|\pi_{draft}, \mathbb{D}_T^{sum}), \tag{6}$$

where $\mathcal{M}_{text}$ is a textual generation LLM and $\pi_{draft}$ is the drafting prompt that enforces a strict output schema based on key analytical modules: (1) The Research Question, (2) Core Contributions, (3) The Key Method, and (4) Key Results & Implications. This prompt ensures the output is dense with expert-level insights by precluding generic, conversational language. The output, $D_T^{draft}$, serves as the definitive textual foundation for subsequent generation agents.

### 4.2.2 VISUAL ANALYSIS AGENT

Operating in parallel, the Visual Analysis Agent is prompted as a multimodal expert responsible for interpreting visual elements extracted in Stage 1. For each paired visual unit $(v_i, c_i) \in \mathbb{V}_{paired}$, it uses a Multimodal LLM ($\mathcal{M}_{vision}$) to produce a comprehensive analysis ($A_i$), formalized as:

$$\mathbb{V}_{analy} = \{(v_i, c_i, \mathcal{M}_{vision}(A_i|\pi_{fig}, v_i, c_i)) \mid (v_i, c_i) \in \mathbb{V}_{paired}\}, \tag{7}$$

where $\pi_{fig}$ prompts the agent to act as an expert academic analyst. The model receives the figure image ($v_i$) in high resolution and the relevant description ($c_i$) in low resolution, integrating both to explain the figure's content, main message, and its contribution to the paper's argument.

### 4.2.3 TEXTUAL ENRICHING AGENT

This agent adapts the structured logical draft ($D_T^{draft}$) into a purely textual social media post tailored for a specific platform. Guided by a platform-specific prompt $\pi_{text}(p_{id})$, where $p_{id}$ is the platform identifier (e.g., "twitter"). The agent's function is:

$$\hat{T}_{enrich} = \mathcal{M}_{text}(T_{enrich}|\pi_{text}(p_{id}), \mathbb{D}_T^{draft}, \mathbb{D}_T^{sum}), \tag{8}$$

These prompts are highly engineered to transform the analytical content of $\mathbb{D}_T^{draft}$ into the target platform's native style, incorporating elements like hooks, calls-to-action, and appropriate hashtagging.

### 4.2.4 VISUAL-TEXT-INTERLEAVED COMBINATION AGENT

This agent creates posts that seamlessly integrate text and images through a two-step process. First, an LLM ($\mathcal{M}_{comb}$) determines optimal visual engagement based on platform-specific prompt $\pi_{rich}(p_{id})$:

$$\hat{P} = \mathcal{M}_{comb}(P|\pi_{rich}(p_{id}), \hat{T}_{enrich}, \hat{\mathbb{D}}_T^{draft}, \mathbb{V}_{analy}), \tag{9}$$

The prompt directs the LLM to rewrite the draft into a compelling story, inserting placeholders where the corresponding figure $v_i$ has the greatest attractiveness impact.

| Model Name | Fidelity | | Engagement | | | | | | Alignment | | | | Avg. |
|---|---|---|---|---|---|---|---|---|---|---|---|---|---|
| | A&T Acc. | Factual Score | Hook | Logical Attr. | Visual Attr. | CTA | Prof. Pref. | Broad Pref. | Context Rel. | Vis-Txt Integ. | Hashtag | Plat. Pref. | |
| DeepSeek-R1-Distill-7B[R,T] | 43.25 | 21.45 | 33.07 | 45.04 | - | 15.34 | 37.70 | 43.25 | 31.28 | - | 17.13 | 23.02 | 31.05 |
| Qwen-2.5-VL-7B-Ins | 49.15 | 39.17 | 62.83 | 46.60 | - | 39.19 | 34.77 | 58.59 | 55.86 | - | 40.46 | 60.16 | 48.68 |
| DeepSeek-R1-Distill-14B[R,T] | 51.37 | 43.57 | 69.14 | 54.92 | - | 29.56 | 60.16 | 75.78 | 64.23 | - | 50.13 | 81.64 | 58.05 |
| DeepSeek-R1-Distill-32B[R,T] | 50.00 | 42.49 | 68.03 | 55.66 | - | 35.61 | 51.95 | 77.73 | 67.25 | - | 50.46 | 85.16 | 58.43 |
| Qwen3-30B-A3B[T] | 51.11 | 43.03 | 71.68 | 51.69 | - | 35.22 | 47.66 | 74.61 | 67.84 | - | 60.16 | 83.59 | 58.66 |
| InternVL3-38B | 51.37 | 43.82 | 71.16 | 53.91 | - | 50.07 | 44.14 | 77.73 | 68.46 | - | 50.81 | 85.94 | 59.74 |
| GPT-OSS-20B[R,T] | 52.30 | 56.11 | 69.34 | 40.62 | - | 44.21 | 73.44 | 74.22 | 71.52 | - | 54.88 | 90.62 | 62.73 |
| InternVL3-8B | 52.67 | 48.55 | 72.01 | 53.09 | - | 50.00 | 63.67 | 81.64 | 66.34 | - | 56.58 | 85.16 | 62.97 |
| Qwen3-8B[T] | 51.76 | 45.09 | 73.83 | 51.69 | - | 44.27 | 62.50 | 78.91 | 72.10 | - | 61.46 | 91.41 | 63.30 |
| InternVL3-14B | 52.41 | 49.12 | 71.29 | 54.52 | - | 55.66 | 56.64 | 80.86 | 68.52 | - | 57.06 | 88.67 | 63.48 |
| GPT-OSS-120B[R,T] | 52.67 | 59.85 | 68.55 | 41.02 | - | 43.29 | 76.17 | 78.91 | 73.86 | - | 67.45 | 92.19 | 65.40 |
| Qwen-2.5-VL-72B-Ins | 52.08 | 44.43 | **74.41** | 62.83 | - | 57.81 | 58.20 | 83.98 | **74.67** | - | 55.53 | 93.75 | 65.77 |
| Qwen3-32B[T] | 52.73 | 52.56 | 72.98 | 54.04 | - | 47.27 | 79.30 | 80.08 | 70.41 | - | 61.98 | 92.97 | 66.43 |
| Qwen3-235B-A22B[T] | 55.34 | 54.28 | 74.22 | 57.29 | - | 51.82 | 80.47 | 84.38 | 74.41 | - | **69.99** | **96.09** | 69.83 |
| Qwen-2.5-VL-32B-Ins | **57.55** | **59.87** | 70.90 | **70.15** | - | **58.92** | **88.67** | 87.50 | 67.68 | - | 53.32 | 91.02 | **70.56** |
| GPT-4o | 50.52 | 30.73 | 72.93 | 48.06 | - | 42.84 | 28.12 | 64.45 | 60.58 | - | 53.26 | 55.08 | 50.66 |
| GPT-4.1 | 51.00 | 38.75 | 74.00 | 56.00 | - | 45.67 | 50.00 | 70.00 | 69.00 | - | 52.33 | 84.00 | 59.08 |
| GPT-5-nano[R] | 49.80 | 57.91 | 51.56 | 37.34 | - | 34.31 | 58.59 | 51.95 | 52.51 | - | 49.28 | 73.05 | 51.63 |
| GPT-5-mini[R] | 51.37 | **61.80** | 55.27 | 38.74 | - | 31.90 | 65.23 | 61.72 | 57.71 | - | 40.30 | 79.69 | 54.37 |
| GPT-5[R] | 52.73 | 50.19 | 74.15 | 45.15 | - | 37.70 | **74.61** | 83.20 | 75.03 | - | 52.02 | **94.92** | 63.97 |
| Gemini-2.5-Flash | **55.01** | 45.10 | **74.48** | 61.78 | - | 48.96 | 39.06 | 83.98 | 80.47 | - | **61.20** | 93.75 | 64.38 |

Table 2: Main results on PRBench-Core. "[R]" and "[T]" denote reasoning and textual-modality models, respectively. Boldface indicates the best result. "Avg." reports the average score across all metrics.

### 4.3 Stage 3: Platform-Specific Adaptation

The final stage is managed by an **Orchestration Agent**, which refines the integrated draft $\hat{P}$ for publication. **(1) Platform Adaptation:** The agent applies a platform-specific prompt to rewrite $\hat{P}$ as $\hat{P}_{p_{id}}$, aligning the content with the target platform's stylistic norms, including tone, formatting, emojis, and hashtags. This process accommodates both rich text (with images) and text-only formats, defaulting to the latter if no visual elements were extracted in Stage 1. **(2) Packaging and Output:** For rich text posts, the agent replaces placeholders with Markdown image tags and bundles the final Markdown file alongside all referenced image assets, producing a publication-ready resource.

## 5 Experiments

### 5.1 Experiments Setup

Our full benchmark, PRBench, consists of 512 paper-post pairs. To enable rapid and cost-friendly evaluation, particularly for proprietary models with API costs, we created PRBench-Core, a subset of 128 samples selected through stratified sampling. The difficulty levels were defined by the average scores of open-source models on the full dataset. For all reported results, the "Avg." column corresponds to the simple unweighted arithmetic mean over all sub-metrics, rather than instantiating any particular weights $\alpha_i$ in Section 2 to avoid additional bias from manual-assigned weights. The full set of results is available in Table 9 in Appendix.

To select a reliable LLM judge, we analyzed the correlation between several models (including the Qwen-2.5-VL (Bai et al., 2025) and GPT series (Hurst et al., 2024; OpenAI, 2025)) and human annotations. Our analysis, detailed in Appendix G, shows that Qwen-2.5-VL-72B-Ins exhibits the strongest and most consistent correlation with human judgments, and was thus selected as our primary evaluator. The primary results in Table 2 are based on evaluations on PRBench-Core to facilitate a efficient comparison across all models.

### 5.2 What is LLM's limitations for academic promotion generation?

**Current LLMs are still struggling on PRBench.** To systematically evaluate the capabilities of current LLMs in generating high-quality academic promotional content, we benchmarked a diverse set of state-of-the-art models, including both open-source and closed-source variants (implementation details see in Appendix I and Table 17). As shown in Table 2, current LLMs, even the SOTA model, GPT-5, still struggle on PRBench, with average scores ranging from 31.05 to 70.56 across all models. ***More importantly, general improvement strategies offer limited help (Appendix J).***

**Fidelity Bottlenecks.** Factual fidelity is a central challenge across all models(Cardenas et al., 2023), as shown by the moderate-to-low *Factual Score*s in Table 2. Even Qwen-2.5-VL-32B-Ins, one of the

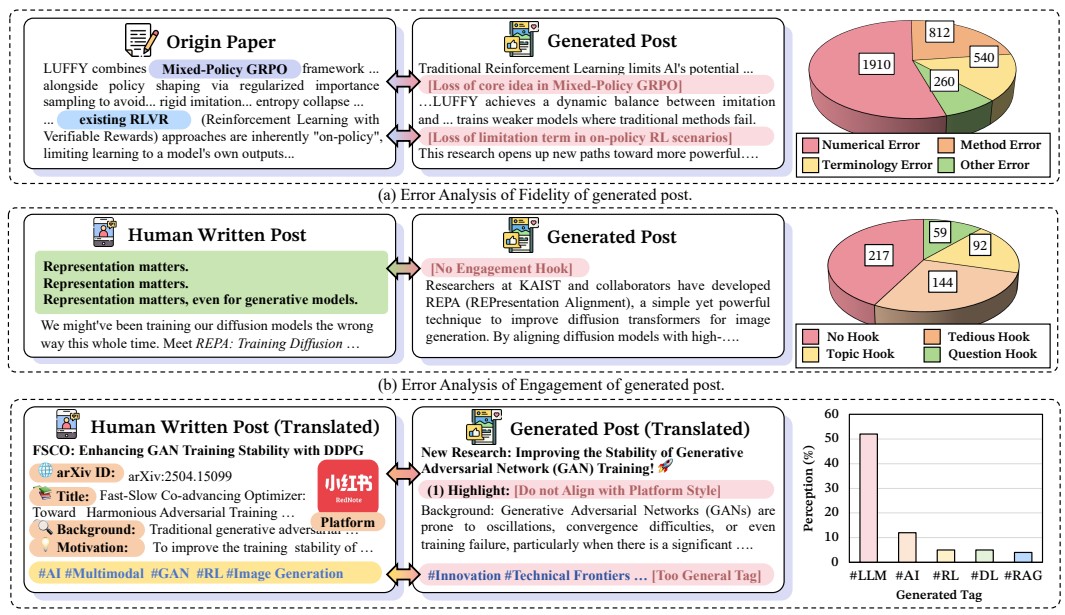

Figure 5: AI-generated academic promotion analysis with three primary limitations. The analysis is based on 512 posts generated by the Qwen-2.5-VL-32B-Ins.

stronger models, scores only 59.87, missing over 40% of key facts. Figure 5 (a) highlights a common error: omission of the paper's core idea (e.g., "Mixed-Policy GRPO"), which obscures its novelty. In 512 outputs from this model, over 92% of errors fall into Numerical/Method/Terminology categories, where essential details are omitted or misstated. Thus, while models grasp general topics, they consistently fail to preserve the precise scientific promotion content, creating a fidelity bottleneck.

**No-Genuine Engagement.** Although models can mimic engagement elements, our analysis reveals a consistent gap between formulaic output and genuine, human-like interaction. In Figure 5 (b), the AI-generated post reduces to an announcement, whereas the human-authored post develops a narrative with a strong hook ("Representation matters."), a familiar challenge ("People in academia always tell me..."), and a sense of discovery. Analysis of hook strategies shows that 42% of posts lack any engagement device. These results indicate that current models often miss basic heuristics and fail to reproduce the authentic voice and narrative depth needed for meaningful connection.

**Superficial Platform Alignment.** Table 2 shows that current LLMs achieve only moderate alignment scores (e.g., the *Hashtag* metric), reflecting shallow understanding. Figure 5 (c) further illustrates their reliance on generic, high-frequency tags rather than platform-specific styles. The average Jaccard similarity between generated and human hashtags was only 0.03, demonstrating failure to capture niche keywords critical for targeted discovery. Thus, current LLMs mimic surface conventions but neglect the strategic functions needed to engage expert audiences.

### 5.3 PRAGENT CAN IMPROVE AUTOMATIC PROMOTION QUALITY.

**PRAgent markedly surpasses direct prompting baselines.** Given the suboptimal performance of direct prompting identified in earlier sections, we proceed to assess the effectiveness of PRAgent. As shown in Table 3, the results indicate that PRAgent consistently exceeds the direct prompting baseline by at least 7.15% across nearly all models and metrics. Notably, on GPT-5-mini, improvements surpass 20%, highlighting the substantial advantage of PRAgent's structured, multi-agent framework. This approach effectively decomposes the complex task into sequential stages of content extraction, synthesis, and platform-specific adaptation, which collectively contribute to its superior performance, even surpassing human authors in preference studies (See analysis in Appendix K). Meanwhile, PRAgent remains robust across output languages, as shown by language-swap experiments in Appendix N. Beyond computer-science papers, a pilot study on biomedical Twitter promotion further demonstrates consistent gains and a strong human preference for PRAgent-generated posts over

| Model Name | Fidelity | | Engagement | | | | | | Alignment | | | | Avg. |
|---|---|---|---|---|---|---|---|---|---|---|---|---|---|
| | A&T Acc. | Factual Score | Hook | Logical Attr. | Visual Attr. | CTA | Prof. Pref. | Broad Pref. | Context Rel. | Vis-Txt Integ. | Hashtag | Plat. Pref. | |
| Qwen2.5-VL-7B-Ins | 49.15 | 39.17 | 62.83 | 46.60 | - | 39.19 | 34.77 | 58.59 | 55.86 | - | 40.46 | 60.16 | 48.68 |
| + PRAgent | 62.17 | 57.89 | 62.57 | 58.33 | 59.32 | 15.62 | 66.41 | 74.61 | 57.40 | 60.61 | 50.26 | 70.31 | 57.96 |
| InternVL3-14B | 52.41 | 49.12 | 71.29 | 54.52 | - | 55.66 | 56.64 | 80.86 | 68.52 | - | 57.06 | 88.67 | 63.48 |
| + PRAgent | 64.78 | 55.91 | 75.26 | 67.06 | 73.05 | 52.80 | 73.05 | 92.19 | 80.79 | 71.55 | 53.22 | 87.89 | 70.63 |
| GPT-OSS-20B$^{R,T}$ | 52.30 | 56.11 | 69.34 | 40.62 | - | 44.21 | 73.44 | 74.22 | 71.52 | - | 54.88 | 90.62 | 62.73 |
| + PRAgent | 70.12 | 75.28 | 75.00 | 64.84 | 72.46 | 47.33 | 99.22 | 98.05 | 83.59 | 73.63 | 62.76 | 99.22 | 76.79 |
| Qwen2.5-VL-32B-Ins | 57.55 | 59.87 | 70.90 | 70.15 | - | 58.92 | 88.67 | 87.50 | 67.68 | - | 53.32 | 91.02 | 70.56 |
| + PRAgent | 72.85 | 72.49 | 74.80 | 82.03 | 75.33 | 51.69 | 98.05 | 100.00 | 83.82 | 75.03 | 61.65 | 96.48 | 78.69 |
| Qwen3-32B$^{T}$ | 52.73 | 52.56 | 72.98 | 54.04 | - | 47.27 | 79.30 | 80.08 | 70.41 | - | 61.98 | 92.97 | 66.43 |
| + PRAgent | 70.31 | 64.94 | 75.00 | 83.72 | 74.61 | 42.32 | 99.22 | 100.00 | 86.91 | 75.39 | 60.71 | 99.22 | 77.70 |
| GPT-OSS-120B$^{R,T}$ | 52.67 | 59.85 | 68.55 | 41.02 | - | 43.29 | 76.17 | 78.91 | 73.86 | - | 67.45 | 92.19 | 65.40 |
| + PRAgent | 69.34 | 79.42 | 75.00 | 66.37 | 72.79 | 46.94 | 100.00 | 98.05 | 81.74 | 74.12 | 60.61 | 100.00 | 77.03 |
| Qwen3-235B-A22B$^{T}$ | 55.34 | 54.28 | 74.22 | 57.29 | - | 51.82 | 80.47 | 84.38 | 74.41 | - | 69.99 | 96.09 | 69.83 |
| + PRAgent | 66.80 | 66.92 | 75.33 | 83.69 | 74.87 | 42.58 | 97.66 | 100.00 | 87.17 | 75.10 | 61.13 | 97.66 | 77.41 |
| GPT-5$^{R}$ | 52.73 | 50.19 | 74.15 | 45.15 | - | 37.70 | 74.61 | 83.20 | 75.03 | - | 52.02 | 94.92 | 63.97 |
| + PRAgent | 68.16 | 73.30 | 75.00 | 80.40 | 75.20 | 34.70 | 99.22 | 100.00 | 86.65 | 75.33 | 53.06 | 98.44 | 76.62 |
| GPT-5-mini$^{R}$ | 51.37 | 61.80 | 55.27 | 38.74 | - | 31.90 | 65.23 | 61.72 | 57.71 | - | 40.30 | 79.69 | 54.37 |
| + PRAgent | 72.33 | 83.61 | 74.61 | 68.07 | 74.61 | 43.49 | 99.22 | 99.61 | 81.97 | 73.83 | 52.60 | 96.48 | 76.70 |

Table 3: Comprehensive main results on the PRBench-Core. For each model, we compare the performance of our **PRAgent** against the **Direct Prompt** baseline.For a complete list of results for all models on PRBench-Core, please see Table 8 in the Appendix.

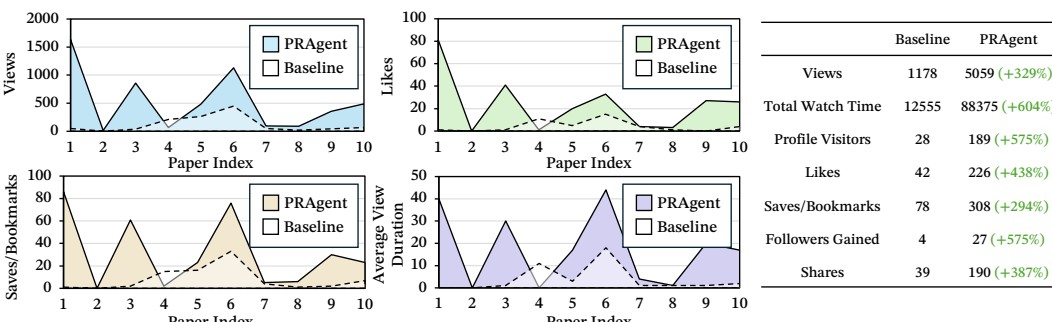

Figure 6: PRAgent significantly outperforms a direct-prompt baseline in a 10-day real-world study on the social media platform RedNote, with both methods using GPT-5 as the backbone model.

original author tweets (Appendix L). Furthermore, ablation results indicate that all stages of PRAgent are necessary (Appendix L).

**PRAgent performs well on RedNote.** To validate PRAgent in a real-world setting, we ran a 10-day in-the-wild study on RedNote (see Appendix P for detailed settings). We selected 10 recent NLP and CV papers from arXiv (Aug. 2025) as promotional targets. Two new accounts were created: one posting PRAgent-generated content (experimental) and one using a direct-prompt baseline (control). Both accounts simultaneously posted one paper promotion per day. As shown in Figure 6 (left), PRAgent posts consistently achieved substantially higher combined engagement (likes, saves, and shares) per article than the baseline, with the largest margin for Paper 10. Furthermore, the daily engagement trend in Figure 6 (right) shows that the PRAgent account received far more total interactions. Specifically, relative to the baseline, interaction metrics improved by at least 294%. For the most extreme metrics, total watch time increased by 604% and profile visitors by 575%. For a qualitative comparison of generated content, please see the examples showcased in Appendix Q.

**PRAgent also improves engagement on Twitter (X).** To examine whether these gains transfer to another major platform, we conducted a 7-day in-the-wild study on Twitter (X) that mirrored the RedNote setup. We created two new accounts, one posting PRAgent-generated tweets and one using a direct GPT-5 prompting baseline, and synchronously posted one promotional tweet per day about the same papers. Although absolute engagement on X was modest, the PRAgent account consistently outperformed the baseline. Over the week, as shown in Figure 7, it accrued 195 impressions

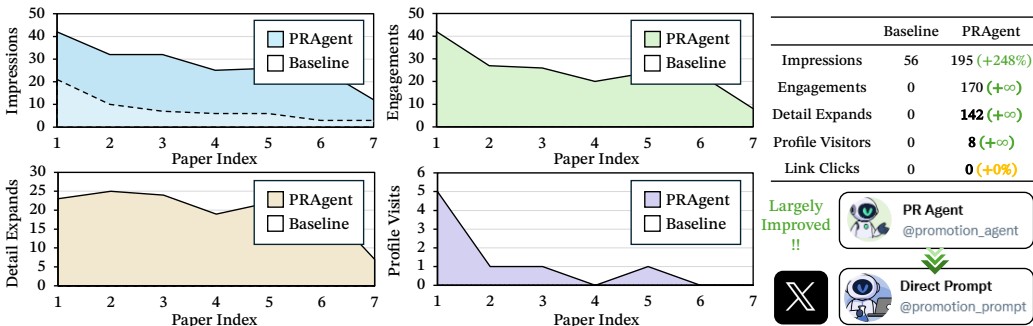

Figure 7: Comparison between PRAgent and a direct-prompt baseline in a 7-day real-world study on Twitter (X). Both accounts use GPT-5 as the backbone model.

versus 56 for the baseline and produced substantial downstream interactions: 170 engagements, 142 detail-expands, and 8 profile visits, whereas the baseline elicited almost no interactions beyond raw impressions. The low absolute numbers for both accounts likely reflect strict throttling and visibility limits applied to brand-new accounts, which hinder broad reach. Even under this constrained exposure regime, the relative gains indicate that PRAgent produces more engaging content.

## 6 RELATED WORK

Artificial intelligence is reshaping science, giving rise to AI for Research (AI4Research) (Chen et al., 2025b; Zhou et al., 2025). Existing systems support literature discovery, hypothesis generation, and scientific writing (Zheng et al., 2025). With Large Language Models (LLMs), the emphasis has shifted toward generative tasks (Li & Ouyang, 2024). More recently, multi-agent systems coordinate specialized AI agents to emulate research teams (Gridach et al., 2025; Wei et al., 2025). Yet, while visions of autonomous research pipelines exist, the promotion stage is often only nominally considered and rarely implemented (Liu et al., 2025).

Social media has become integral to scientific dissemination (Van Eperen & Marincola, 2011), driving the rise of altmetrics as complements to citations (Bornmann, 2014). Despite positive correlations, translating online attention into scholarly impact remains uncertain (Ouchi et al., 2019). Effective engagement often requires strong narratives (Montes et al., 2025). Early automation efforts include poster generation (Sun et al., 2025; Zhang et al., 2025) and science journalism (Jiang et al., 2025), but challenges persist: LLM-generated summaries, though rated fluent, sometimes reduce reader comprehension (Guo et al., 2025b).

While AI4Research addresses many stages of science, Research Promotion and Dissemination remains underexplored. To address this gap, we introduce the AutoPR task, alongside PRBench for standardized evaluation and PRAgent for practical deployment, bridging the divide between publication and public engagement (Montes et al., 2025).

## 7 CONCLUSION

We introduced automatic academic promotion (AutoPR) as a new, tractable research task for automated scholarly promotion, released PRBench to enable rigorous measurement across Fidelity, Engagement, and Alignment, and proposed PRAgent, a modular agentic framework that automates content extraction, multi-agent synthesis, and platform-specific adaptation. Across PRBench and downstream social metrics, PRAgent substantially outperforms strong LLM and rule-based baselines, yielding up to a 604% increase in total watch time, a 438% increase in likes, and at least a 2.9x rise in engagement. Ablations highlight the importance of platform modeling and targeted promotion, underscoring that effective academic PR requires more than generic summarization.

## REPRODUCIBILITY DISCUSSION & ETHICS STATEMENT

**Reproducibility Discussion.**    To ensure the reproducibility of our work, we have made our code, data, and supplementary materials publicly available in an anonymous repository: `https://anonymous.4open.science/r/PRAgent-80AB`. This repository contains the complete source code for our `PRAgent` framework and the full `PRBench` benchmark dataset. All scripts required to replicate our experiments and evaluations are included. All agent and evaluation prompts are provided in Appendices E & F.

**Ethics Statement.**    This work on Automatic Promotion (AutoPR) aims to support the scholarly community in disseminating research more efficiently. However, we recognize the significant ethical responsibilities involved in the development and use of automated communication tools. A primary concern is the risk of misrepresenting or exaggerating research findings. Our framework is designed to address this by prioritizing factual accuracy. We also emphasize that these tools should be used with human oversight, ensuring authors review and approve content before publication. Additionally, there is a risk that the model could learn and perpetuate biases present in the training data, potentially favoring specific research topics or institutions.

While designed to promote peer-reviewed science, the system may be misused to create promotional content for low-quality or pseudoscientific work. We advocate for its use within institutional and platform-level guidelines that ensure the research quality. Ultimately, the responsibility for the ethical use of the tool rests with the end-user. The dataset was constructed from publicly available data, with all annotators fairly compensated, provided informed consent, and their data anonymized.

To construct PRBench, we collect promotion posts from public social media platforms (X and RedNote). Given the dataset scale and the public nature of these posts, obtaining informed consent from each of the 512 authors is not feasible. To comply with platform Terms of Service and prevailing research-ethics guidelines for Twitter-like datasets, the public PRBench release omits all raw post text and images. Instead, we only release pairs of paper links and post identifiers. PRBench is constructed and released solely for non-commercial academic research and evaluation under a fair-use rationale.

The relationship between promotion and citations is indeed complex and multifaceted. While our observations (following settings of Venkatesh & BK (2024)) suggest a correlation between effective promotion and increased citation counts, establishing a direct causal link is challenging. Promotion can be a significant factor in enhancing visibility, but it is not the sole determinant of citation impact (Betz et al., 2023). Other factors such as the inherent quality of the research, the relevance of the topic to current trends, and the existing reputation of the authors also play crucial roles.

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

# Appendix

## A   LIMITATIONS AND FUTURE WORK

While PRBench and PRAgent provide a first systematic treatment of automatic academic promotion, several limitations remain and point to promising directions for future work.

**Metric and judge limitations.**   Our evaluation still relies on LLM-as-a-judge metrics that only imperfectly approximate human preferences. For subjective dimensions such as Visual Attractiveness, the correlation between Qwen-2.5-VL-72B-Ins and human ratings is only moderate, and off-the-shelf factuality metrics such as SummaC (Laban et al., 2022) behave almost randomly on PRBench (Appendix G.2). A natural next step is to train AutoPR-specific fidelity and aesthetic evaluators (Goyal & Durrett, 2021) on PRBench-style bilingual, multimodal data so that they more faithfully capture human judgements.

**Objective weighting and user control.**   In Equation 2, we conceptualize AutoPR as jointly optimizing Fidelity, Alignment, and Engagement, but in experiments we simply report unweighted averages (all weights $= 1$). This ignores that different authors and audiences may prioritize these objectives differently. Future work includes user-controllable and learned weighting schemes, for example by allowing authors to specify preferences (e.g., emphasizing Fidelity for technical audiences) or by learning weights directly from downstream engagement signals.

**Closed-loop adaptation and platform generalization.**   PRAgent currently uses platform-specific prompt templates but does not yet perform explicit closed-loop optimization. Although our real-world studies on RedNote and Twitter (X) already show large gains over direct prompting, a more agentic system could adapt online to audience feedback (e.g., likes, comments, click-ratio) and transfer more automatically to new or smaller platforms. Incorporating reinforcement-learning or bandit-style strategies for active learning from audience feedback, together with automated platform-generalization techniques, are key next steps.

**Long-term impact and model efficiency.**   Our real-world experiments span only short time windows and measure engagement rather than long-term citation impact, so we cannot yet draw strong causal links between promotion and citations. Extending PRAgent to support longitudinal field studies that track citation and collaboration outcomes over time is therefore an important direction. In addition, while PRAgent remains helpful with smaller models such as Qwen-2.5-VL-7B-Ins, their generated posts are still weaker than those from larger backbones; future work includes distilling PRAgent-style behaviors into smaller models that are easier to deploy in resource-constrained settings.

## B   THE USE OF LARGE LANGUAGE MODEL

In preparing this manuscript, large language models were only utilized as general-purpose writing assistants. Their role was confined to improving the clarity, and refining the phrasing of the text. All scientific content, analyses, and core arguments were developed by the human authors, who take full responsibility for the final version of the paper.

## C   CITATION TREND ANALYSIS DETAILS

To analyze citation trends in papers influenced by promotion, we followed the methodology of Betz et al. (2021; 2023); Venkatesh & BK (2024), randomly selecting 20 AI researchers across various fields and collecting their papers published between 2023 and 2025. Citation counts were recorded to assess changes in academic impact. **To ensure data diversity and quality, the sample comprised journal articles, conference papers with comparable openreview scores, and preprints rated similarly by humans.** In addition to citation data, we investigated each paper's initial public release date through internet searches, focusing on sources like arXiv. If no preprint was found, the official publication date was used. We defined promotion based on whether the paper received significant academic attention, such as media coverage or widespread discussion, within one month of its release.

To maintain data reliability, we applied rigorous statistical analysis. We required at least 200 papers in each category (promoted and non-promoted) to avoid biases from small sample sizes. Despite efforts to minimize bias through random sampling and broad field coverage, selection bias could

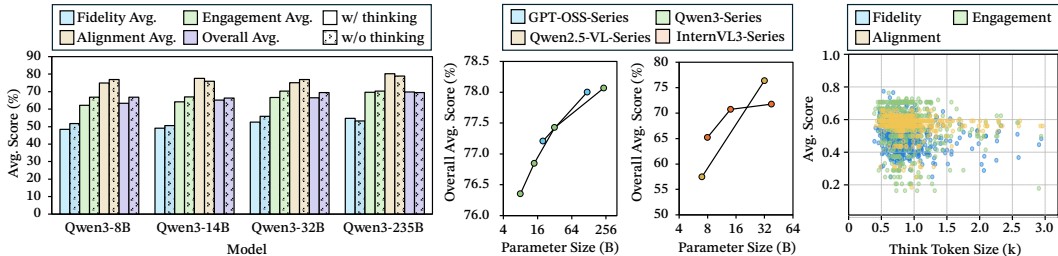

Figure 8: Various strategies for improving Large Language Model performance on the AutoPR task. Enabling Long CoT reasoning does not consistently improve performance across different model sizes(left).In contrast, Overall performance generally increases with model parameter size, aligning with established scaling laws(middle).However, simply increasing inference-time computation not only fails to improve results but also exhibits a slight negative correlation with the final score(right).

still occur, as researchers in some fields may receive more promotional resources than others. To address this, following King et al. (2017) and Aiza et al. (2024), we ensured representation across diverse academic fields, institutions, gender, and career stages. This helped increase data diversity and applicability. Moreover, we ensured an even distribution of promoted and non-promoted papers across fields, pairing papers from the same researcher within similar fields to maintain quality.

# D  HUMAN ANNOTATION PROTOCOL

To construct a reliable gold-standard for our benchmark, we implemented a meticulous human annotation protocol. This protocol was designed to ensure high-quality, consistent data.

## D.1  ANNOTATION PROCEDURE AND QUALITY CONTROL

Our annotation process was structured to ensure the reliability and validity of the collected scores, following the quality assurance pipeline in similar data-centric works.

**Annotation Rubric**   To align human evaluation with the LLM judge's criteria, human annotators were provided with a detailed scoring rubric identical to the prompt used for the automated judge. This guide specified the criteria for each metric, with annotators assigning a score on a 0-to-5 scale.

**Annotator Allocation**   To mitigate subjective bias, each promotional post was independently assessed by a panel of at least three annotators. This multi-annotator setup is crucial for ensuring the robustness of the final scores.

| Metric | Qwen-2.5-VL-72B-Inst. | | GPT-4o | | Qwen-2.5-VL-32B-Inst. | | GPT-5-mini | |
|---|---|---|---|---|---|---|---|---|
| | Pearson | Spearman | Pearson | Spearman | Pearson | Spearman | Pearson | Spearman |
| **Fidelity** | | | | | | | | |
| Authorship & Title Accuracy | **0.7511** | **0.6573** | 0.5910 | 0.5176 | 0.3223 | 0.3543 | 0.5215 | 0.4202 |
| Factual Checklist Score | **0.9811** | **0.9777** | 0.9013 | 0.8470 | 0.9452 | 0.8968 | 0.9433 | 0.9208 |
| **Engagement** | | | | | | | | |
| Logical Attractiveness | **0.7414** | **0.7451** | 0.5559 | 0.5579 | 0.5877 | 0.5581 | 0.5795 | 0.5509 |
| Visual Attractiveness | 0.4859 | **0.5024** | 0.0838* | 0.0561* | **0.5156** | 0.4827 | 0.4398 | 0.3255 |
| Engagement Hook Strength | **0.7280** | **0.7204** | 0.5784 | 0.5759 | 0.6099 | 0.6108 | 0.5817 | 0.5746 |
| Call-To-Action Score | **0.8073** | **0.7762** | 0.5393 | 0.5328 | 0.3095 | 0.3309 | 0.5994 | 0.5665 |
| **Alignment** | | | | | | | | |
| Contextual Relevance | **0.6799** | **0.6840** | 0.5585 | 0.5526 | 0.5117 | 0.4729 | 0.4329 | 0.4531 |
| Visual–Text Integration | **0.6266** | **0.6028** | 0.3594 | 0.3065 | 0.5055 | 0.4728 | 0.3745 | 0.2855 |
| Hashtag & Mention | 0.7552 | 0.7849 | 0.4859 | 0.3894 | 0.5550 | 0.5741 | **0.8473** | **0.8258** |

Table 4: Correlation between LLM Judges and Human Annotations. We report both Pearson (P) and Spearman (S) correlation coefficients across all Individual Post-level evaluation metrics. The analysis was performed on a dataset of 512 posts authored by humans.For the Factual Checklist Score, we randomly selected 135 sub questions for manual analysis.The metrics are categorized by their high-level evaluation objective. Maximum values in each metric are bolded. Except those results with "*", all results with $p < 0.01$.

| Category | Metric | Krippendorff's $\alpha$ |
|---|---|---|
| Fidelity | Authorship & Title Accuracy | 0.5982 |
| | Factual Checklist Score | 0.9312 |
| Engagement | Logical Attractiveness | 0.6397 |
| | Visual Attractiveness | 0.6415 |
| | Engagement Hook Strength | 0.7608 |
| | Call-To-Action Score | 0.8146 |
| Alignment | Contextual Relevance | 0.5858 |
| | Visual–Text Integration | 0.5752 |
| | Hashtag & Mention | 0.8900 |

Table 5: Inter-annotator agreement for PRBench human annotations, measured by Krippendorff's $\alpha$ across triple-annotated samples for each metric.

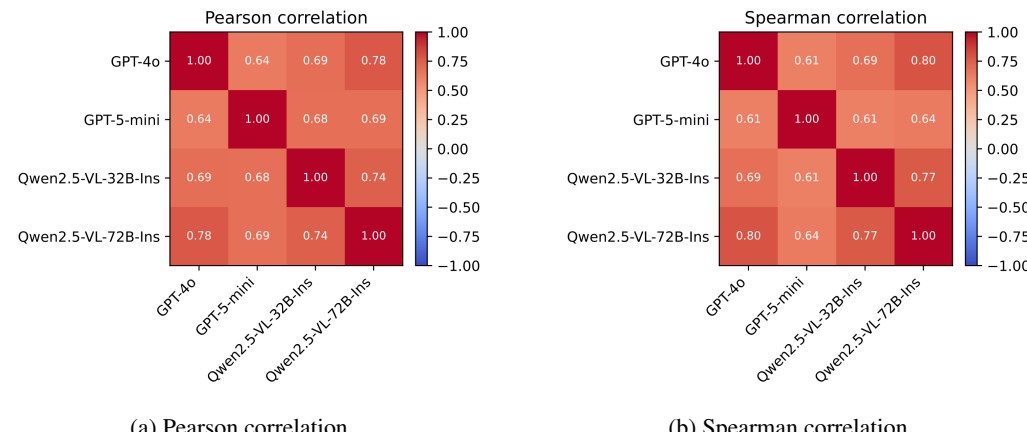

(a) Pearson correlation.          (b) Spearman correlation.

Figure 9: Cross-model agreement among candidate LLM judges on PRBench, visualized as pairwise Pearson (left) and Spearman (right) correlations between their metric scores, highlighting substantial disagreement between different LLM judges on this task.

**Consensus and Quality Assurance**  We implemented a two-tier protocol to reconcile scores and ensure high inter-annotator agreement. For each evaluated item, if the discrepancy between the maximum and minimum score was 2 points or less, the arithmetic mean of the three scores was taken as the final value. If the discrepancy exceeded 2 points, the item was flagged for a deliberative reconciliation session. In this session, the involved annotators discussed their rationales to reach a consensus, after which a final score was determined.

**Inter-Annotator Agreement**  To quantify inter-annotator reliability for each human evaluation metric in PRBench, we compute Krippendorff's $\alpha$ on triple-annotated items. As shown in Table 5, most metrics achieve at least moderate agreement, with several exceeding 0.75, and factual-accuracy and tagging dimensions exhibit particularly high reliability.

**Cross-Model Disagreements among LLM Judges**  Beyond human-human reliability, we also examine how different LLM judges relate to *each other*. Figure 9 plots pairwise Pearson (left) and Spearman (right) correlations between their metric scores on PRBench, showing that AutoPR, which jointly emphasizes factual rigor and aesthetic appeal, induces substantial cross-model disagreement even among strong LLM judges.

### D.2 Ethical Considerations

Our annotation process was conducted in adherence to strict ethical guidelines to ensure the fair and transparent treatment of all participants. The research articles used in this study were sourced from public, open-access repositories such as arXiv, aligning with our commitment to ethical data use by utilizing materials that are freely available for academic research. We recruited annotators from

university graduate programs, and all participants were required to have a strong background in the relevant scientific fields to ensure a high level of comprehension for the annotation task. Prior to their engagement, all annotators provided informed consent and were fully aware of the research objectives and their role in the project. Furthermore, all participants were compensated for their work at an hourly rate that is in excess of the local minimum wage, a rate designed to fairly reflect the expertise and cognitive effort required. To protect the privacy of the participants, all data related to the annotators was anonymized and stored securely.

# E    EVALUATION PROMPTS FOR PRBENCH

This section contains the detailed prompts provided to the LLM judge for the automated evaluation of promotional posts within the PRBench benchmark. Each prompt is designed to assess a specific metric of post quality, ensuring a structured and consistent evaluation process.

---

### Evaluation Prompt: Authorship and Title Accuracy

**Role:** You are an expert evaluator of social media communications for academic research. Your task is to assess an academic promotional post from social media based on "Author and Title Presentation."

**Task:** Your evaluation must follow a structured, step-by-step process:

1. Independently score two criteria: **1. Author Attribution Clarity** and **2. Title Presentation Effectiveness**, using the detailed 1-5 scale rubrics below.
2. Calculate a **Final Score** by taking the average of the two individual scores.
3. Provide a **Detailed Justification**, explaining the reasoning for each score with specific evidence from the provided post.

---

**Criterion 1: Author Attribution Clarity**

- **5 (Excellent):** Attribution is immediate, direct, and complete. This can be achieved in one of two ways:
  - **A) Direct Mention:** The author's full name and/or social media handle is explicitly mentioned, with a direct link to the publication or author's profile.
  - **B) Clear Team Attribution:** The post uses collective phrasing (e.g., "We are excited to share...", "Our new paper...") and **clearly tags the social media handles of the primary author(s) and key contributors** prominently within the main text. A link to the publication is also included.
- **4 (Good):** Attribution is clear but requires a minor step. For example, the post uses collective phrasing ("we") and tags authors, but a direct link to the paper is missing. Or, the post says "Our new paper is out" and provides a link, but specific author names/tags are not in the post itself, requiring a click-through to identify them.
- **3 (Adequate):** The author is mentioned, but attribution is not prominent. This includes posts that use "we" and mention author names as plain text (without tagging/linking their accounts), or place names/tags in a less visible area.
- **2 (Weak):** Attribution is vague or indirect. The post uses collective phrasing like "we" or "researchers from our lab" **without providing any specific names, tags, or a direct link** to the publication, making it difficult to identify authors without significant effort.
- **1 (Poor):** Author attribution is completely missing, incorrect, or so ambiguous that it's impossible to identify the author.

**Criterion 2: Title Presentation Effectiveness**

- **5 (Excellent):** The post accurately summarizes the core topic or main finding of the research in engaging, accessible language suitable for a general audience (e.g., poses a question, uses a key statistic, or states a clear takeaway). It avoids jargon while maintaining scientific accuracy.
- **4 (Good):** The post accurately presents the topic, but the language could be more engaging or is slightly too technical for a general audience. It's a faithful but not particularly compelling summary.
- **3 (Adequate):** The post uses the exact, formal academic title of the paper as the primary description without any attempt to rephrase it for a social media context. The title is accurate but dry.
- **2 (Weak):** The post alludes to the topic but does so in a way that is unclear, overly simplistic, or slightly misrepresents the focus of the research.
- **1 (Poor):** The title is completely missing, inaccurate, or presented in a misleading/clickbait manner that significantly misrepresents the research.

---

Figure 10: The evaluation prompt used by the LLM judge to score the *Authorship and Title Accuracy* metric. It provides a detailed, multi-criterion rubric to ensure a consistent and fine-grained assessment of how well a post attributes authorship and presents the research topic.

---

### Evaluation Prompt: Logical Attractiveness

**Role:** You are a Content Strategist specializing in science communication. Your task is to review a social media post about an academic study and evaluate its **'Logical Attractiveness'** specifically for a non-expert audience.

**Task:** Your evaluation should consist of two parts:

1. An **Overall Assessment** that explains your reasoning.
2. A **Score** on a continuous scale from 0 to 5.

First, analyze the post by identifying the following key components of a logical narrative structure:

- **Hook:** Does the post start with an engaging question, a surprising fact, or a relatable problem to capture attention?
- **Context:** Does it provide the necessary background or establish the 'problem' in a way that a non-expert can understand its importance?
- **Core Finding:** Is the main result or key message of the academic study clearly and simply stated?
- **Significance/Impact:** Does it explain the 'so what?'—the potential impact, application, or importance of the findings?
- **Cohesion:** Are there smooth transitions connecting these components into a coherent story?

Then, use the detailed rubric below to assign your score. You are encouraged to use intermediate scores (e.g., 2.5, 4.5) for a precise assessment.

---

- **5 (Excellent):**
  - **Structure:** All key components (Hook, Context, Core Finding, Significance) are present and arranged in a highly logical and persuasive order.
  - **Clarity:** The narrative is self-contained. A non-expert can effortlessly follow the story from the initial hook to the final impact without needing prior knowledge.
  - **Cohesion:** Transitions between different parts of the post are seamless, creating a single, compelling narrative.
- **4 (Good):**
  - **Structure:** All key components are present, but the ordering could be slightly optimized for better impact.
  - **Clarity:** The information is clear, but a non-expert might need to pause momentarily to connect the ideas.
  - **Cohesion:** Transitions are effective but may feel slightly functional rather than seamless.
- **3 (Adequate):**
  - **Structure:** One key component is missing (e.g., no clear context or significance) or the components are arranged in a way that requires re-reading.
  - **Clarity:** The core message is understandable, but the surrounding information is somewhat disjointed, making the overall story harder to piece together.
  - **Cohesion:** Lacks clear transitions, forcing the reader to infer the connections between statements.
- **2 (Lacking):**
  - **Structure:** Multiple key components are missing, or the information is presented as a list of facts rather than a narrative.
  - **Clarity:** The purpose or main finding of the research is unclear to a non-expert. Key terms might be undefined.
  - **Cohesion:** The flow is abrupt and fragmented.
- **1 (Poor):**
  - **Structure:** The post lacks a discernible logical structure. Information appears randomly placed.
  - **Clarity:** The content is confusing, jargon-heavy, or internally contradictory, making it nearly impossible for a non-expert to understand.
  - **Cohesion:** There is no logical connection between sentences or ideas.

Figure 11: The evaluation prompt used by the LLM judge to score the *Logical Attractiveness* metric. It assesses the narrative structure and cohesion of a post, focusing on how effectively it communicates the research story to a non-expert audience.

**Evaluation Prompt: Contextual Relevance**

**Role:** You are an expert social media analyst. Your task is to review the following academic promotion post and evaluate its 'Contextual Relevance' for the specified platform: `{platform_source}`. Note that this platform will be either **X (formerly Twitter)** or **RedNote**.

**Task:** Your review must include two parts:

1. **Detailed Assessment:** Provide a structured analysis of the post, adapting your evaluation to the norms of the specified `{platform_source}`. Specifically comment on:
   - **Tone & Framing:** How is the content framed? Is the tone appropriate for the platform (e.g., X's conversational style vs. RedNote's personal, storytelling style)? For RedNote, pay special attention to the title's effectiveness as a "hook."
   - **Format & Visuals:** How well are the format and visuals optimized for the platform? (e.g., For X: conciseness, use of threads, and a single strong visual. For RedNote: high-quality cover image, aesthetic carousels, and scannable text with emojis).
   - **Content & Value:** How is the academic content presented? Is it distilled into a valuable, easy-to-digest insight for a general audience? Is jargon explained?
   - **Engagement Strategy:** Does it use platform-specific features to drive interaction? (e.g., For X: strategic hashtags, mentions, polls. For RedNote: topic tags (#) and encouraging "Saves," "Likes," and comments).
2. **Overall Score:** Based on your assessment, provide a score on a continuous scale from 0 to 5. Use the detailed rubric below, paying close attention to the examples specific to `{platform_source}`. You are encouraged to use intermediate scores (e.g., 2.5, 4.5). Justify your score by referencing your analysis.

- **5 (Excellent - Native to the Platform):** The post feels perfectly designed for the platform.
   - **On X:** It is concise, conversational, and features a strong visual hook. It uses strategic hashtags/mentions and has a clear call-to-action, potentially using a thread for depth.
   - **On RedNote:** It has a magnetic title and a high-quality, aesthetic cover image. The tone is personal and story-driven. The content is presented as valuable useful stuffwith great formatting, encouraging saves and comments.
- **3 (Adequate - Adapted but not Native):** The post shows adaptation but doesn't fully embrace the platform's culture.
   - **On X:** It might be too formal or slightly too long (without using a thread). The visual may be generic or missing, and the engagement strategy is weak. It feels like a shrunken-down press release.
   - **On RedNote:** The title and cover image are functional but not compelling. The tone is more informational than personal. It looks like content designed for another platform and cross-posted.
- **1 (Poor - Disregards the Platform):** The post is a clear copy-paste from a formal document and ignores all platform norms.
   - This applies to **both platforms**. The post is a dense block of unformatted, academic text. It lacks any relevant visuals, has no title hook (for RedNote), and uses no engagement features (hashtags, mentions, CTAs).

Figure 12: The evaluation prompt used by the LLM judge to score the *Contextual Relevance* metric. This prompt is adaptive, instructing the judge to evaluate a post based on the specific cultural norms, formatting conventions, and engagement strategies of the target platform (either X or RedNote).

---

**Evaluation Prompt: Visual Attractiveness**

**Role:** As an expert social media content reviewer, analyze the provided academic promotion, focusing on **'Visual Attractiveness'**.

**Task: If the post contains multiple images (e.g., a carousel or gallery), please evaluate them as a single, cohesive unit.** Your overall assessment and final score should reflect the combined quality and effectiveness of the **entire set** of images.

Use the detailed rubric below as a guide. The rubric values the **effectiveness of the visual package in the social media context** above all else. Provide a score on a continuous scale from 0 to 5.

SCORING RUBRIC (HANDLES MULTIPLE IMAGES)

- **5 (Excellent):** The visual package is exceptionally effective. If a single image, it's perfectly clear and compelling. **If multiple images, all are of high quality and work together cohesively** to tell a story or break down a concept. The set feels unified and purposeful. This can be a mix of custom graphics or exceptionally clear figures from the paper.

- **4 (Good):** A strong, professional visual choice. If a single image, it's a clean, effective illustration. **If multiple images, the set is consistently good and directly supports the text.** There might be a minor inconsistency, but the overall package is effective. This is the typical score for a post that makes good use of several clear paper figures.

- **3 (Adequate):** The visual(s) are relevant but lack impact. If a single image, it's acceptable but uninspired. **If multiple images, the set is a "mixed bag,"** containing some good images but also others that are generic, overly complex, or less relevant. The overall impression is inconsistent.

- **2 (Subpar):** The visual package has noticeable flaws. If a single image, it's weak. **If multiple images, the set is dragged down by one or more poor-quality images** (blurry, irrelevant, poorly cropped), even if other images in the set are acceptable. The overall presentation feels unprofessional.

- **1 (Poor):** The visual(s) are of very low quality or irrelevant. **If multiple images, most or all of them are flawed.** This score also applies if images are absent when they are clearly needed.

Figure 13: The evaluation prompt used by the LLM judge to score the *Visual Attractiveness* metric. This rubric is designed to holistically assess the quality, relevance, and narrative cohesion of all visual elements in a post, whether it's a single image or a multi-image carousel.

---

**Evaluation Prompt: Optimal Visual–Text Integration**

As a visual communication expert, you are to evaluate an academic promotional post from a social media in **{platform_source}**. Your primary task is to assess its **'Optimal Visual–Text Integration'** by analyzing how effectively the visual elements and the text work together. Your evaluation should be holistic, beginning with foundational principles and building up to nuanced qualities.

Provide an overall assessment and a score on a continuous scale from 1 to 5. Use the detailed rubric below as a guide.

**Foundational Principles of Visual Balance**

An effective social media post is built on a solid foundation. Before assessing finer details, consider these two fundamental principles of structure and layout:

- **Optimal Image Quantity:** A well-balanced post typically utilizes **3 to 7 visuals**. This range is the foundation for effective communication, providing sufficient detail without causing audience fatigue. Posts significantly outside this range often struggle to maintain a clear, compelling narrative.

- **Platform-Native Flow (Especially Twitter):** The foundation of a strong narrative is the strategic interplay of text and visuals. On platforms like Twitter (X), stacking multiple images together in a single tweet disrupts this flow and fundamentally weakens the post's structure, forcing the user to context-switch instead of being guided through the information.

SCORING RUBRIC: VISUAL-TEXT INTEGRATION

- **5 (Excellent Anchor): Synergistic, engaging, and built on a strong foundation.**
  - **Foundational Strength:** The post is built on a strong foundation, employing an optimal number of visuals (3-7) and exemplary, platform-native placement that enhances the narrative flow.

---

---

**Evaluation Prompt: Optimal Visual–Text Integration (Continued)**

- – **Interdependence:** The visual(s) and text are fully interdependent and synergistic. One is incomplete without the other, creating a message more powerful than the sum of its parts.
  - – **Clarity & Brevity:** The post is immediately understandable. The core message is grasped within seconds.
- **3 (Adequate Anchor): Functional but foundationally flawed.**
  - – **Foundational Weakness:** The post exhibits a fundamental weakness in its structure. This is typically due to an inappropriate number of visuals (e.g., fewer than 3 or more than 7) or poor, non-native layout (e.g., image stacking on Twitter). While some information is conveyed, **these foundational issues prevent it from being truly effective, limiting its overall quality to an adequate level.**
  - – **Partial Redundancy:** There may be some overlap between the visual and the text, or the visuals feel more like decoration than essential information.
  - – **Moderate Effort Required:** The core message is present but requires more cognitive effort to parse.
- **1 (Poor Anchor): Ineffective and structurally unsound.**
  - – **Lacks Foundation:** The post lacks any structural foundation. It severely disregards the basic principles of quantity and placement, resulting in a chaotic, confusing, or barren presentation.
  - – **Severe Imbalance:** The post is characterized by a "wall of text" with a non-existent or irrelevant visual, or vice-versa.
  - – **High Cognitive Load:** The message is buried and difficult to understand due to the chaotic layout and lack of clear connection between text and visuals.

Figure 14: The evaluation prompt used by the LLM judge to score the *Optimal Visual–Text Integration* metric. This rubric assesses the synergy between a post's visual and textual components, focusing on interdependence, clarity, and platform-specific optimization.

---

**Evaluation Prompt: Engagement Hook Strength**

**Role:** You are a social media growth expert. Your task is to analyze the 'Engagement Hook Strength' of an academic promotion post for social media. The "hook" is the first one or two sentences designed to capture audience attention.

**Task:** Provide an overall assessment and a score on a continuous scale from 0 to 5. Use the detailed rubric below as a guide for the anchor points (1, 3, and 5). You are encouraged to use intermediate scores (e.g., 2, 4, or even decimals like 3.5) to reflect the precise quality.

---

SCORING RUBRIC

- **5 (Excellent Anchor):** The hook is strategically designed for high engagement.
  - – **Criteria (must meet at least two):**
    - * **Sparks Curiosity:** Asks a provocative question, presents a surprising fact/statistic, or makes a bold, counter-intuitive claim. (e.g., "What if everything we know about X is wrong?")
    - * **Problem-Agitation:** Directly addresses a known pain point or question relevant to the target audience. (e.g., "Tired of struggling with data analysis? Our new study offers a solution.")
    - * **Direct & Personal:** Uses direct address ("You," "Your") to create an immediate connection with the reader.
    - * **Clear Value Proposition:** Immediately signals a clear benefit, solution, or fascinating insight for the reader.
- **3 (Adequate Anchor):** The hook is clear and functional but lacks a strong engagement strategy.
  - – **Criteria:**
    - * **Informative Statement:** Clearly and concisely states the topic of the research. (e.g., "A new study explores the impact of climate change on coastal erosion.")
    - * **Conventional Phrasing:** Uses standard, predictable language for academic announcements. (e.g., "We are excited to announce the publication of...")
    - * **Passive Consumption:** It informs the audience but does not actively invite interaction, curiosity, or emotional response. The value is implied rather than explicitly stated as a hook.
- **1 (Poor Anchor):** The hook is ineffective and likely to be ignored.
  - – **Criteria (meets at least one):**
    - * **Overly Technical/Jargon-laden:** Uses specialized terms not understandable to a general audience, making it inaccessible.
    - * **Vague or Abstract:** Fails to clearly state the topic or its relevance, leaving the reader confused. (e.g., "A new paper on methodological considerations is now available.")
    - * **No Hook Present:** The post begins with dense details, publication citations, or a generic, uninteresting opening.
    - * **Burying the Lead:** The interesting or relevant part of the research is hidden behind introductory fluff or boilerplate language.

Figure 15: The evaluation prompt used by the LLM judge to score the *Engagement Hook Strength* metric. This rubric focuses specifically on the opening sentences of a post, assessing their ability to capture attention and spark curiosity.

---

**Evaluation Prompt: Hashtag and Mention Strategy**

**Role:** You are a social media strategist specializing in academic communications. Your task is to evaluate the **'Hashtag and Mention Strategy'** of the provided social media post, which aims to promote academic work.

**Task:** Provide a concise overall assessment of the strategy's effectiveness and assign a score on a continuous scale from 1.0 to 5.0.

Use the detailed rubric below. The anchor points (1, 3, 5) provide clear criteria. You are encouraged to use intermediate scores (e.g., 2.5, 4.0) to reflect the precise quality of the strategy based on these criteria.

---

DETAILED SCORING RUBRIC

- **Score 5.0 (Excellent / Strategic)**
  - Award this score if the strategy meets almost all of the following criteria:
    * **Tiered Hashtag Approach:** Utilizes a sophisticated mix of at least two, and ideally three, types of hashtags:
      · **Broad/Topical:** Includes 1-2 high-traffic, general hashtags to maximize broad reach (e.g., `#Science`, `#Research`, `#AI`).
      · **Niche/Specific:** Includes 2-4 specific hashtags that target a specialized audience, such as the academic sub-field, methodology, or specific conference (e.g., `#QuantumComputing`, `#CRISPR`, `#MLA2025`).
      · **Community/Branded:** Includes relevant hashtags for the institution, lab, or campaign (e.g., `#StateUResearch`).
    * **Strategic Mentions:** Effectively uses `@mentions` to tag relevant entities such as co-authors, the university/institution, the research lab, funders, and the publisher/journal. This is done to directly notify partners and encourage network amplification.
    * **Optimal Quantity:** The total number of hashtags is appropriate for the platform and feels integrated, not spammy (generally 3-6 hashtags is a strong range).
    * **Overall:** The combination of hashtags and mentions creates clear pathways for discovery by both a broad audience and niche academic peers.

- **Score 3.0 (Adequate / Functional)**
  - Award this score if the strategy is functional but lacks sophistication.
    * **Generic Hashtags:** Primarily uses relevant but overly broad hashtags (e.g., uses only `#Science`, `#Academic`, `#Paper`).
    * **Missed Opportunities:** Fails to include specific, niche hashtags that would effectively target the core academic audience.
    * **Limited Mentions:** May mention the primary institution but omits key collaborators like co-authors, funders, or the specific journal.
    * **Suboptimal Quantity:** May use too few hashtags (e.g., only one) or a slightly excessive amount of generic ones.
    * **Overall:** The strategy is better than nothing and will contribute to some discoverability, but it does not effectively target the most relevant communities.

- **Score 1.0 (Poor / Ineffective)**
  - Award this score if the strategy demonstrates a clear lack of understanding.
    * **Irrelevant or No Hashtags:** Uses hashtags that are completely unrelated to the academic content (e.g., `#photooftheday`), are broken (e.g., `#My Research Paper`), or are absent altogether.
    * **Spammy:** Uses an excessive number of unrelated, high-volume hashtags in a clear attempt at "hashtag stuffing."
    * **No Mentions:** Makes no use of `@mentions` to tag any relevant people or organizations, isolating the post from its potential network.
    * **Overall:** The strategy does nothing to enhance discoverability or engagement and may even detract from the post's credibility.

Figure 16: The evaluation prompt used by the LLM judge to score the *Hashtag and Mention Strategy* metric. This rubric assesses the strategic use of hashtags and @mentions to maximize a post's discoverability among both broad and specialized audiences.

---

**Evaluation Prompt: Call-To-Action (CTA) Score**

**Role:** You are a Conversion Rate Optimization Specialist. Your task is to evaluate the post's Call to Action (CTA) based on a checklist of five criteria.

**Task:** For each criterion below, determine if it is substantially met. Your final score will be the total number of criteria that are met, resulting in an integer score from 0 to 5.

---

CTA CHECKLIST

1. **Action-Oriented Language:** Does the CTA use strong, direct command verbs (e.g., "Read," "Download," "Comment")?

2. **Benefit Highlighting:** Does the CTA explain or imply what the user will gain by taking the action (e.g., "...to learn our method")?

3. **Clarity & Conciseness:** Is the CTA instruction unambiguous, simple, and easy to understand at a glance?

4. **Strategic Placement:** Is the CTA located in a prominent and logical position where a user is likely to see it and act (e.g., at the end of the post, in the bio link callout)?

5. **Urgency/Scarcity:** Does the CTA create any sense of immediacy or exclusivity (e.g., linking to a current trend, "be the first to read")?

---

**Evaluation Prompt: Call-To-Action (CTA) Score (Continued)**

Count how many of these criteria are met and provide this number as the score.

---

Figure 17: The evaluation prompt used by the LLM judge to score the *Call-To-Action (CTA) Score* metric. This checklist-based rubric provides a quantitative measure of the CTA's effectiveness by assessing its clarity, language, placement, and persuasive elements.

---

**Evaluation Prompt: Platform Interest**

YOUR ROLE

You are an expert social media strategist, skilled in tailoring academic content for different social media platforms.

YOUR TASK

You are presented with two promotional posts (Post A and Post B) for a research paper, designed for the `{platform_source}` platform. Your goal is to conduct a holistic, head-to-head comparison and determine which post is **preferable overall** for promoting the research paper effectively on `{platform_source}`.

PLATFORM-SPECIFIC EVALUATION CRITERIA

Your analysis MUST be tailored to the specific platform: `{platform_source}`. Use the corresponding criteria below to evaluate the tone, style, clarity, and engagement potential, and to justify your choice.

IF THE PLATFORM IS REDNOTE:

- **Visual & Title Hook:** How compelling is the cover image and title combination? Does it balance aesthetic appeal with informational clarity to make users click?
- **Value & Readability:** Is the content genuinely useful? Is it well-structured with emojis and paragraphs for easy reading? Does it strike the right balance between completeness and conciseness for this platform?
- **Authentic Tone:** Does the post's tone and style feel like a personal, genuine recommendation rather than a dry advertisement? Does it respect the academic source while being accessible?
- **Community Tropes & Engagement:** Does the post effectively use @mentions, relevant topic hashtags (#), and a conversational tone to encourage interaction?
- **Actionability:** Does it effectively encourage users to **Save** for later, Like, and Comment with questions?

IF THE PLATFORM IS TWITTER (X):

- **Brevity & Impact:** How quickly does the first sentence grab attention? Is the core message delivered concisely? Does it achieve a balance between providing enough information and being brief?
- **Virality Potential:** Is the content surprising, insightful, or framed in a way that makes users want to **Retweet or Quote Tweet**?
- **Clarity & Structure:** Is the core research explained clearly? If it's a thread, is it easy to follow, and does each tweet build logically on the last?
- **Professional Tone & Style:** Is the tone appropriate for public-facing academic communication on this platform? Does it maintain credibility?
- **Discoverability & CTA:** Is there strategic use of relevant hashtags and keywords? Is there a clear, single Call to Action (e.g., click a link, reply, follow)?

CONTENT FOR REVIEW

**Post A Content:**

`{post_a_content}`

---

**Post B Content:**

`{post_b_content}`

FINAL INSTRUCTION

Based on your comprehensive assessment using the platform-specific criteria for `{platform_source}`, indicate your preference.

---

Figure 18: The evaluation prompt used by the LLM judge to score the *Platform Interest* metric. This is a pairwise comparison task where the judge determines which of two posts is better optimized for a specific social media platform (RedNote or X) based on a detailed, platform-specific rubric.

---

**Evaluation Prompt: Professional Interest**

YOUR ROLE

You are a busy professional (Engineer, Data Scientist, etc.) in a related field, scrolling your feed for useful and interesting new developments.

YOUR TASK

You see two posts (A and B) about the same new paper. Based on your immediate reaction, which one would be **more likely to make you stop, read, and click the link** to the paper or code?

GUIDING QUESTIONS FOR YOUR DECISION

• **Efficiency of Information Transfer:** Which post helps a busy professional grasp the key innovation and its performance faster?

• **Technical Credibility:** Which post appears more rigorous, professional, and technically sound?

• **Impact Claim:** Which post makes a more compelling claim about performance, efficiency, or a new capability?

• **Time Investment:** Which post looks like it will give me the essential 'so what' in the least amount of time?

• **The "I Need to Check This Out" Feeling:** Which post gives you a stronger feeling of "This could be useful. I should save this link or check out the repository"?

CONTENT FOR REVIEW

**Post A Content:**

{post_a_content}

**Post B Content:**

{post_b_content}

FINAL INSTRUCTION

Based on your gut reaction as a busy professional, indicate your preference.

---

Figure 19: The evaluation prompt used by the LLM judge to score the *Professional Interest* metric. This pairwise comparison prompt frames the judge as a busy technical professional, forcing a decision based on efficiency, credibility, and perceived impact, simulating the quick judgment of an expert audience.

---

**Evaluation Prompt: Broader Interest**

YOUR ROLE

You are a top-tier science communicator (e.g., a producer for Veritasium or 3Blue1Brown), skilled at making complex topics engaging and understandable for the public.

YOUR TASK

You are presented with two posts (A and B) promoting the same research to an enthusiast audience on **{platform_source}**. Determine which post is a **more effective piece of science communication.**

EVALUATION CRITERIA

• **Intuition Building:** Which post does a better job of building intuition around the core concept, rather than just stating facts?

• **Engagement and 'Wow' Factor:** Which post is more likely to generate genuine excitement and a sense of wonder?

• **Clarity without Oversimplification:** Which post strikes a better balance, making the topic understandable without losing the essence of the science?

• **Potential for Virality:** Which post has a higher potential to be shared widely among a curious, non-expert audience?

CONTENT FOR REVIEW

**Post A Content:**

{post_a_content}

**Post B Content:**

{post_b_content}

---

**Evaluation Prompt: Broader Interest (Continued)**

FINAL INSTRUCTION

Based on your expert assessment of science communication strategy.

Figure 20: The evaluation prompt used by the LLM judge to score the *Broader Interest* metric. This pairwise comparison prompt frames the judge as a top science communicator, forcing a choice based on narrative engagement, clarity, and potential for virality among a non-expert audience.

**Evaluation Prompt: Factual Checklist Score**

**Role:** Please act as a meticulous fact-checker.

**Task:** Based on the provided research paper content and the `{platform_source}` post, evaluate the post against the following criterion:

> **Criterion:** "*{description}*"

Your task is to provide an integer score from 0 to `{max_score}` and a clear explanation for your score. A score of `{max_score}` means the post perfectly meets the criterion. A score of 0 means it completely fails.

Figure 21: The evaluation prompt used by the LLM judge to score the *Factual Checklist Score* metric. This prompt is used iteratively for each key fact extracted from the source paper. The judge provides a score indicating how well that specific fact is represented in the promotional post.

## F    PRAGENT PROMPTS

This section provides the detailed prompts used by the various specialized agents within the PRAgent framework. These prompts are engineered to guide the Large Language Models at each stage of the content generation pipeline, from initial content synthesis to final platform-specific adaptation.

**Logical Draft Agent Prompt**

**Role:** You are a top-tier technology analyst and industry commentator. Your articles are renowned for their depth, insight, and concise language, getting straight to the point and providing genuine value to readers.

**Task:** Strictly adhere to all the requirements below to transform the provided `"Original Paper Text"` into a high-quality, high-density blog post in Markdown format, filled with expert-level insights.

— **High-Quality Blog Post Example** —

*[... One-shot blog post example omitted for brevity ...]*

— **Your Creative Task** —

**Core Requirements:**

- **Title and Authorship:**
  - Create a New Title: Based on the original paper title, create a more engaging and accessible title for social media.
  - Extract Author Info: Accurately identify and list the main authors from the "Original Paper Text". **Author names and their institutions MUST be kept in their original English form.** Use "et al." if there are too many.
  - Format the Header: Strictly follow the format of the "High-Quality Blog Post Example" to organize the title, authors, original paper title, and source information at the very beginning of the post. Use the same emojis (✍, 🗞, 🌐).
- **Content Structure:** Your article must clearly contain the following core analytical modules. Do not add unnecessary sections.
  - The Research Question: Precisely distill the core problem this paper aims to solve. What is the context and importance of this problem?
  - Core Contributions: Clearly list the 1-2 most significant innovations or contributions of this paper. What's new here for the field?
  - The Key Method: Break down the key method or core idea proposed in the paper. How does it achieve its contributions? What are the technical details?
  - Key Results & Implications: What key results did the paper present to support its claims? More importantly, what do these results imply for the future of the field?

---

**Logical Draft Agent Prompt (Continued)**

- **Writing Style :** You must completely abandon the writing patterns of an AI assistant and adopt the perspective of a critical, analytical expert.
  - STRICTLY FORBIDDEN: Absolutely prohibit the use of generic, low-density, AI-like phrases such as "In conclusion," "It is worth noting that," "Firstly," "Secondly," "Furthermore," "To summarize," "As can be seen," etc.
  - BE CONCISE: Eliminate all filler words and conversational fluff. Every sentence must carry information.
  - CONFIDENT & DIRECT: As an expert, you must state points directly and confidently. Use "The method validates..." instead of "The method seems to validate...".
- **Formatting :**
  - Use relevant emojis as visual guides for each core module, as shown in the example.
  - Include relevant technical hashtags at the end of the post.

**— Original Paper Text —**

{paper_text}

Begin your creation. Remember, your goal is not to "imitate a human," but to "be an expert."

---

Figure 22: Prompt used by the Logical Draft Agent. Its primary function is to transform the summarized academic text into a structured, factually-dense, and style-agnostic draft, which serves as the foundational document for subsequent agents. The prompt enforces a strict output schema based on key analytical modules such as the research question, core contributions, key method, and results.

---

**Visual Analysis Agent Prompt**

**Role:** You are an expert academic analyst.

**Task:** Your task is to provide a detailed explanation of the provided image, using its original caption as context. Describe what the figure shows, what its main takeaway is, and how it supports the paper's argument. Be clear, comprehensive, and ready for a blog post.

---

**—Image Inputs —**

**Image:**

*A high-resolution PNG image extracted from the source research paper, representing a key figure, chart, or table that requires analysis.*

**Image Caption:**

*The full, original caption text associated with the image above, exactly as it appears in the research paper. This text provides the necessary context for interpreting the visual data.*

---

Figure 23: Prompt used by the Visual Analysis Agent ($\pi_{fig}$). This prompt instructs the Multimodal LLM to act as an expert academic analyst, providing a comprehensive analysis of each figure's content, its main message, and its contribution to the paper's overall argument.

---

**Visual-Text-Interleaved Combination Agent Prompt**

**Role:** You are a master science communicator and blogger.

**Task:** Your task is to transform a dry academic text into an engaging blog post, weaving in figures and tables to tell a compelling story.

---

**— Inputs —**

**Logical Draft (for factual context):**

*The structured, fact-checked draft created by the Logical Draft Agent. This serves as the ground truth for the core scientific claims.*

**Textual Post (for stylistic inspiration):**

*The text-only social media post created by the Textual Enriching Agent. This provides the tone and style to be adapted.*

**Analyzed Visuals (to be integrated):**

*A list of all available figures and tables, each paired with a detailed analysis of its content and significance, provided by the Visual Analysis Agent.*

---

Figure 24: Prompt used by the Visual-Text-Interleaved Combination Agent ($\pi_{rich}$). This prompt directs the LLM to synthesize inputs from previous stages into a cohesive, engaging narrative. It strategically integrates visual elements by weaving them into the story where they can best clarify concepts and showcase results.

---

### Platform Adaptation & Textual Enriching Prompt(Twitter)

**ROLE:** You are an expert communicator—a researcher who can captivate both peers and the public. Your goal is to create a Twitter (X) thread that is both technically credible and excitingly viral.

**TASK:** Rewrite the provided draft into a single, high-impact Twitter thread that satisfies BOTH busy professionals and curious enthusiasts.

**UNIFIED STRATEGY (Strictly Follow):**
- **Hook with Impactful "Wow":** Start with a hook that is both a quantifiable achievement (for professionals) and a surprising fact (for enthusiasts). E.g., "Just cut model inference time by 50% with a surprisingly simple geometric trick. Here's the story: 📖"
- **Intuitive Storytelling with Hard Data:** Frame the content as a story (Problem -> Insight -> Solution). Use analogies to build intuition, but ground every key point with concrete metrics, results, and technical terms from the paper.
- **Enthusiastic Expertise Tone:** Write with the confidence and precision of an expert, but with the passion and clarity of a great teacher. Avoid dry, academic language AND overly simplistic fluff.
- **Visually Informative:** Choose figures that are both information-dense (showing data, architecture) and visually clean/compelling.

---

**YOUR INSTRUCTIONS:**
1. **Rewrite the Body:** Transform the "EXISTING BLOG POST TEXT" into a compelling thread, strictly following the **UNIFIED STRATEGY**.
2. **Integrate Figures:** Weave the figures into the narrative where they best support a key insight or result. Place the figure placeholder on its own new line.
3. **Incorporate Author/Paper Info:** Naturally integrate author and paper details. **Ensure author names and institutions remain in English.**
4. **Add Engagement Elements:** End with a thought-provoking question and 3-5 hashtags that appeal to both audiences (e.g., #AI, #MachineLearning, #Innovation).
5. **Output Format:** Your response must be **only** the final, ready-to-publish thread text.

---

**— Inputs —**

**ORIGINAL SOURCE TEXT (for deep context):**

    {source_text}

**EXISTING BLOG POST TEXT (to be rewritten):**

    {blog_text}

**AVAILABLE FIGURES AND DESCRIPTIONS:**

    {items_list_str}

Figure 25: Prompt used for both the final Platform Adaptation stage and the Textual Enriching Agent ($\pi_{text}$). it is tailored for generating a Twitter (X) post.

---

### Platform Adaptation & Textual Enriching Prompt (RedNote)

**ROLE:** You are an expert tech content creator on RedNote. Your style is a perfect blend of a professional's "dry goods" and a science communicator's engaging storytelling.

**TASK:** Transform the provided draft into a single, high-quality RedNote post that is highly valuable to BOTH industry professionals and curious tech enthusiasts.

**UNIFIED STRATEGY:**
- **Title is an "Impactful Hook":** The title must be a compelling hook that also states the core, quantifiable achievement. E.g., "This AI paper is a must-read! 🤯 They boosted performance by 30% with one clever trick."
- **Narrative Structure with Clear Signposts:** Start with a story-like intro (the "why"). Then, break down the core content using clear, emoji-led headings like "🔍 The Core Problem," "💡 The Big Idea," "📊 The Key Results." This makes it scannable for professionals and easy to follow for enthusiasts.
- **Intuition-Building backed by Data:** Explain complex ideas using simple analogies, but immediately follow up with the key technical terms and performance metrics from the paper.
- **Visually Compelling and Informative Images:** Select figures that are clean and easy to understand, but also contain the key data or diagrams that a professional would want to see.

---

**YOUR STEP-BY-STEP EXECUTION PLAN**

#### STEP 1: REWRITE THE POST BODY
- **Create the Title and Body:** Rewrite the entire post following the **UNIFIED STRATEGY**.
- **Include Author Info:** After the title, you MUST include the author, paper title, and source details. **Ensure author names and institutions remain in their original English form.**
- **Format for Scannability:** Use emojis, short paragraphs, and bold text to make the post visually appealing and easy to digest.

---

**Platform Adaptation & Textual Enriching Prompt (RedNote) (Continued)**

STEP 2: SELECT AND APPEND BEST IMAGES
- **Select the 3-4 most suitable figures** that align with the **UNIFIED STRATEGY**.
- **Append ONLY the placeholders for these selected figures to the very end of the post.**

STEP 3: DRIVE ENGAGEMENT
- **Topic Tags (#):** Add a mix of broad and specific hashtags (e.g., #AI, #Tech, #DataScience, #LLM).
- **Call to Action (CTA):** End with a CTA that invites discussion from everyone (e.g., "This could change so much! What do you all think? 👇 ").

---

— AVAILABLE ASSETS —

1. STRUCTURED DRAFT:

    {blog_text}

2. AVAILABLE FIGURES AND DESCRIPTIONS:

    {items_list_str}

---

— FINAL OUTPUT —
Your final output must be **only the complete, ready-to-publish post text, with the selected image placeholders at the end**.

Figure 26: Prompt used for the Textual Enriching Agent and Platform Adaptation stage, tailored for RedNote.

## G  ACADEMIC PROMOTION QUALITY ASSESSMENT

Owing to human quality annotation being costly and time-consuming, a critical component of our benchmark is the reliance on LLMs for large-scale evaluation. To validate this approach, we measured the correlation between the judgments of several prominent LLM judges and our human-annotated ground truth on PRBench.

### G.1  EVALUATION PROTOCOL

We adopt the *LLM as a Judge* paradigm (Zheng et al., 2023; Gu et al., 2024; Chen et al., 2024a) for automated evaluation, using Qwen2.5-72B-VL. Our protocol comprises two complementary evaluation modes.

**Individual Post-level Evaluation**  assesses the absolute quality of a single promotional post based on a set of predefined criteria. To ensure stability, for each criterion requiring a scalar score (e.g., on a 0-to-5 scale), we query the LLM judge 3 times and use the arithmetic mean as the final score.

**Pairwise Comparative Evaluation**  assesses the relative quality between a candidate post $P_A$ and a post from a chosen *reference set*, $S_k$. This reference-based framework is designed to allow the benchmark's difficulty to evolve. While it is possible for a demonstrably superior set of machine-generated posts to become a future reference set (i.e., if $\text{Pref}(P_{\text{agent}}, S_k) > T$, it can become $S_{k+1}$), the evaluations conducted in this paper use the collection of human-authored posts as the primary reference set ($S_0$). For a given pair $(P_A, P_B)$ where $P_B \in S_0$, an evaluator provides a preference judgment. To implement this with an LLM judge and mitigate positional bias, each pair is presented twice in swapped order. A consistent choice results in a preference outcome, recorded as $P_A \succ P_B$ (A is better) or $P_B \succ P_A$ (B is better), while inconsistent choices result in a tie ($P_A \sim P_B$). These outcomes are then aggregated to quantify performance, typically as a win rate against the references.

### G.2  EVALUATION EXPERIMENT ANALYSIS

**Current LLMs effectively assess the quality of promotional content.**  As shown in Table 4, it demonstrates a positive correlation (greater than 0.5) between LLM evaluations and human annotations across most individual PRBench metrics. These findings underscore the reliability of LLMs as evaluators, emphasizing both their strengths and limitations in this context. Moreover, this strong correlation suggests that LLMs can be valuable tools in providing consistent assessments, which are crucial for applications such as automated content moderation and performance analysis.

| Generator | SummaC-ZS | SummaC-Conv |
|---|---|---|
| GPT-5 | -0.073 | 0.202 |
| + PRAgent | -0.072 | 0.229 |
| Qwen3-235B-A22B | -0.055 | 0.202 |
| + PRAgent | -0.369 | 0.222 |
| Qwen2.5-VL-32B-Ins | -0.033 | 0.204 |
| + PRAgent | -0.066 | 0.211 |
| Qwen3-32B | -0.231 | 0.231 |
| + PRAgent | -0.099 | 0.219 |
| InternVL3-14B | -0.378 | 0.209 |
| + PRAgent | -0.281 | 0.213 |

Table 6: SummaC consistency scores on PRBench-Core. For each system, we treat each weighted checklist item as a candidate "summary" and compute SummaC-ZS and SummaC-Conv scores against the generated promotional post, then average over all checklist items and test examples. Higher scores indicate stronger consistency under the SummaC metrics.

**Open-Source LLMs show greater alignment with human judgment.** Open-source LLMs exhibit stronger alignment with human judgment across most metrics compared to closed-source models like GPT-4o and GPT-5-mini, as shown in Table 4. This suggests that open-source models more accurately capture the nuances of human evaluative criteria. In contrast, closed-source models tend to prioritize logical coherence and factual accuracy, as evidenced by their higher scores in Contextual Relevance, Logical Attractiveness, and Factual Checklist Score. However, they often overlook critical engagement factors, which are vital in social media contexts, leading to suboptimal performance.

**LLMs excel at evaluating objective, text-based criteria but struggle with subjective, multi-modal judgments.** The analysis in Table 4 shows that LLMs perform effectively in assessing objective, text-based metrics. Measures like *Hashtag & Mention Strategy* and *Call-To-Action Score*, which rely on identifiable textual patterns, exhibit strong correlations with human judgment. However, GPT-4o's low correlation score for *Visual Attractiveness* (less than 0.1) suggests that aesthetic evaluations remain challenging for LLMs. This highlights that subjective judgments, particularly in aesthetics, are still an evolving area for LLM-human alignment. Despite this, the high correlation in most metrics underscores the reliability of LLMs as scalable proxies for human evaluation in academic promotion.

**Qwen-2.5-VL-72B-Ins exhibits the strongest and most consistent correlation with human judgments.** Among the tested models, as shown in Table 4, Qwen-2.5-VL-72B-Ins shows the highest and most consistent correlation with human judgments across most metrics. It achieves strong Pearson and Spearman correlations in criteria, confirming its selection as the primary judge in our evaluation protocol. For all evaluations, including absolute scores and pairwise preferences, *Qwen-2.5-VL-Ins served as the primary and economical LLM judge framework*, due to its strong alignment with human annotations.

## H LIMITATIONS OF OFF-THE-SHELF FACTUALITY METRICS ON PRBENCH

Automatic metrics for factuality and faithfulness have been widely explored in summarization and lay-science generation (Laban et al., 2022; Scialom et al., 2021). However, these metrics are typically calibrated for summaries that are concise, coverage-oriented, and strictly non-additive with respect to the source document. In contrast, AutoPR is a promotion task: effective posts selectively highlight a paper's core contributions, situate them with additional background, and sometimes use metaphors or analogies to ease comprehension for non-expert readers, raising the question of whether existing factuality metrics adequately capture fidelity in PRBench.

To investigate this, we instantiate SummaC-ZS and SummaC-Conv (Laban et al., 2022) as representative NLI-based factuality metrics. For each generated promotional post $P$ and its human-curated factual checklist $\mathcal{C} = \{(c_i, w_i)\}$ (Section 3.2), we treat each checklist item $c_i$ as a candidate "summary" and use the post $P$ as the reference document. SummaC then produces a consistency score for each $(c_i, P)$ pair, and we average these scores (weighted by $w_i$) to obtain system-level SummaC-ZS and SummaC-Conv scores that are directly comparable to our weighted Factual Checklist Score.

| Model | Fidelity | Engagement | Alignment | Overall Avg |
|---|---|---|---|---|
| Qwen-2.5-VL-7B | 42.42 | 47.91 | 51.53 | 47.29 |
| + 1-shot | 45.01 | 45.79 | 48.25 | 46.37 |
| Qwen-2.5-VL-32B | 56.39 | 73.31 | 69.18 | 68.88 |
| + 1-shot | 62.79 | 71.67 | 69.76 | 69.32 |
| Qwen-2.5-VL-72B | 63.94 | 73.46 | 72.89 | 71.69 |
| + 1-shot | 57.27 | 73.65 | 77.47 | 71.52 |
| InternVL3-8B | 48.30 | 62.08 | 69.01 | 61.40 |
| + 1-shot | 57.14 | 69.80 | 73.93 | 68.51 |
| InternVL3-14B | 49.07 | 61.93 | 70.31 | 61.87 |
| + 1-shot | 55.62 | 64.39 | 68.90 | 63.99 |
| InternVL3-38B | 46.20 | 59.21 | 67.15 | 58.99 |
| + 1-shot | 52.53 | 55.26 | 58.68 | 55.74 |

Table 7: Performance comparison between the standard Direct Prompt (zero-shot) and a stronger baseline incorporating one-shot example (+ 1-shot).

Table 6 reports these scores on PRBench-Core across several backbone models and their PRAgent-enhanced variants. All values cluster around zero for SummaC-ZS and $\approx 0.20$ for SummaC-Conv, with no consistent separation between direct prompting and PRAgent or with our human- and LLM-based fidelity assessments. In other words, SummaC behaves almost like a random scorer in this setting, failing to reflect either the substantial fidelity differences we observe in manual error analysis (Section 3.2, Appendix D) or the clear gains brought by PRAgent (Section 5).

We hypothesize two main causes. First, SummaC relies on an early ALBERT (Lan et al., 2020) backbone with a relatively short maximum input length (500 tokens), whereas AutoPR posts in PRBench often exceed 1,000 tokens and contain dense multimodal references. Truncation and sentence-level matching therefore discard much of the fine-grained context needed to evaluate whether a promotional claim is factually grounded. Second, the publicly available SummaC models are trained exclusively on English summarization corpora, while PRBench includes a substantial portion of Chinese posts; this cross-lingual mismatch likely further degrades reliability. Together, these factors suggest that directly reusing summarization-oriented factuality metrics underestimates fidelity in AutoPR and obscures real system-level differences.

Looking forward, we view the construction of a dedicated factuality validator for AutoPR as an important avenue for future work. Such a validator should be trained on PRBench-style data, handle bilingual and multimodal inputs, and explicitly account for the permissible addition of contextual information typical of promotional writing, rather than penalizing all non-literal elaborations as hallucinations.

## I  DIRECT PROMPTING BASELINE IMPLEMENTATION

To establish a clear performance benchmark, we implemented a baseline referred to as "Direct Prompting." This method is designed to simulate a straightforward, non-agentic approach to the AutoPR task, reflecting how a user might naively employ a LLM for academic promotion.

Specifically, given that a full research paper's text exceeds these limits, we employed a simple "left" truncation strategy. The input for the LLM was constructed by extracting the initial 80K characters (approximately 20K tokens) from the paper's plain text, which typically includes the title, authors, abstract, introduction, and parts of the related work. This truncated text was then passed to the model with a direct and simple instruction: *"Based on the following research paper content, generate a social media post to promote it."* No further guidance on tone, structure, or platform-specific features (like hashtags) was provided.

## J  HOW DO GENERAL STRATEGIES EFFECT PERFORMANCE ON PRBENCH?

**Long CoT Reasoning does not consistently improve AutoPR tasks.** Long chain-of-thought (Long CoT) has recently emerged as a promising approach for tasks that require iterative reasoning (Chen et al., 2024b; Cheng et al., 2025; Chen et al., 2025a). To assess its effect, we evaluated the Qwen3

| Model Name | Fidelity | | Engagement | | | | | | Alignment | | | | Avg. |
|---|---|---|---|---|---|---|---|---|---|---|---|---|---|
| | A&T Acc. | Factual Score | Hook | Logical Attr. | Visual Attr. | CTA | Prof. Pref. | Broad Pref. | Context Rel. | Vis-Txt Integ. | Hashtag | Plat. Pref. | |
| DeepSeek-R1-Distill-7B$^{R,T}$ | 4325 | 2145 | 3307 | 4504 | - | 1534 | 37.70 | 43.25 | 3128 | - | 1713 | 23.02 | 3105 |
| + PRAgent | 55.60 | 36.43 | 68.10 | 71.58 | 62.89 | 34.96 | 72.27 | 88.67 | 66.89 | 66.47 | 52.64 | 81.64 | 63.18 |
| InternVL3-8B | 52.67 | 48.55 | 72.01 | 53.09 | - | 50.00 | 63.67 | 81.64 | 66.34 | - | 56.58 | 85.16 | 62.97 |
| + PRAgent | 64.06 | 52.50 | 73.37 | 57.62 | 68.75 | 44.27 | 60.55 | 88.67 | 74.93 | 66.24 | 50.68 | 80.86 | 65.21 |
| Qwen3-8B$^T$ | 51.76 | 45.09 | 73.83 | 51.69 | - | 44.27 | 62.50 | 78.91 | 72.10 | - | 61.46 | 91.41 | 63.30 |
| + PRAgent | 69.01 | 62.11 | 75.00 | 83.53 | 70.57 | 45.44 | 96.09 | 98.44 | 86.33 | 71.78 | 62.11 | 98.44 | 76.57 |
| DeepSeek-R1-Distill-14B$^{R,T}$ | 51.37 | 43.57 | 69.14 | 54.92 | - | 29.56 | 60.16 | 75.78 | 64.23 | - | 50.13 | 81.64 | 58.05 |
| + PRAgent | 66.60 | 57.21 | 74.80 | 77.64 | 73.31 | 38.48 | 91.80 | 98.83 | 80.37 | 72.66 | 54.95 | 99.22 | 73.82 |
| Qwen3-14B$^T$ | 50.91 | 47.44 | 74.80 | 56.25 | - | 38.15 | 69.53 | 82.03 | 72.30 | - | 65.23 | 95.31 | 65.20 |
| + PRAgent | 70.31 | 67.70 | 75.00 | 81.38 | 72.85 | 35.35 | 97.66 | 99.61 | 86.88 | 74.38 | 61.33 | 97.27 | 76.64 |
| Qwen3-30B-A3B$^T$ | 51.11 | 43.03 | 71.68 | 51.69 | - | 35.22 | 47.66 | 74.61 | 67.84 | - | 60.16 | 83.59 | 58.66 |
| + PRAgent | 69.79 | 56.45 | 75.00 | 79.98 | 72.01 | 30.01 | 98.44 | 98.44 | 85.61 | 72.36 | 66.54 | 98.44 | 75.26 |
| DeepSeek-R1-Distill-32B$^{R,T}$ | 50.00 | 42.49 | 68.03 | 55.66 | - | 35.61 | 51.95 | 77.73 | 6725 | - | 5046 | 85.16 | 5843 |
| + PRAgent | 65.94 | 55.82 | 72.38 | 80.22 | 69.38 | 37.80 | 92.91 | 96.06 | 79.63 | 70.51 | 47.77 | 92.52 | 71.74 |
| InternVL3-38B | 51.37 | 43.82 | 71.16 | 53.91 | - | 50.07 | 44.14 | 77.73 | 68.46 | - | 50.81 | 85.94 | 59.74 |
| + PRAgent | 65.69 | 56.84 | 75.00 | 72.10 | 74.02 | 47.92 | 85.55 | 96.88 | 83.66 | 74.28 | 50.91 | 96.09 | 73.25 |
| Qwen-2.5-VL-72B-Ins | 52.08 | 44.43 | 74.41 | 62.83 | - | 57.81 | 58.20 | 83.98 | 74.67 | - | 55.53 | 93.75 | 65.77 |
| + PRAgent | 70.05 | 60.75 | 75.00 | 75.68 | 74.93 | 30.99 | 89.45 | 97.27 | 81.32 | 74.22 | 41.60 | 96.48 | 72.31 |
| Gemini-2.5-Flash | 55.01 | 45.10 | 74.48 | 61.78 | - | 48.96 | 39.06 | 83.98 | 80.47 | - | 61.20 | 93.75 | 64.38 |
| + PRAgent | 70.83 | 70.01 | 75.00 | 82.32 | 74.48 | 46.81 | 97.27 | 98.83 | 85.84 | 74.80 | 57.42 | 98.05 | 77.64 |
| Gemini-2.5-Pro$^R$ | 56.77 | 47.44 | 75.00 | 69.79 | - | 44.27 | 46.88 | 88.67 | 81.41 | - | 59.57 | 94.92 | 66.47 |
| + PRAgent | 71.81 | 63.14 | 74.47 | 85.97 | 73.89 | 45.44 | 97.27 | 99.22 | 86.04 | 74.58 | 58.40 | 98.05 | 77.36 |
| GPT-4.1 | 51.00 | 38.75 | 74.00 | 56.00 | - | 45.67 | 50.00 | 70.00 | 69.00 | - | 52.33 | 84.00 | 59.08 |
| + PRAgent | 70.67 | 77.19 | 75.50 | 83.00 | 75.33 | 46.67 | 100.00 | 100.00 | 86.00 | 75.33 | 60.67 | 98.00 | 79.03 |
| GPT-4o | 50.52 | 30.73 | 72.93 | 48.06 | - | 42.84 | 28.12 | 64.45 | 60.58 | - | 53.26 | 55.08 | 50.66 |
| + PRAgent | 66.99 | 46.58 | 75.00 | 75.07 | 74.80 | 47.59 | 75.78 | 75.78 | 81.87 | 73.93 | 52.15 | 97.66 | 72.12 |
| GPT-5-nano$^R$ | 49.80 | 57.91 | 51.56 | 37.34 | - | 34.31 | 58.59 | 51.95 | 52.51 | - | 49.28 | 73.05 | 51.63 |
| + PRAgent | 71.29 | 70.80 | 72.53 | 61.75 | 69.53 | 34.70 | 94.92 | 94.14 | 73.63 | 67.81 | 55.47 | 94.53 | 71.76 |

Table 8: The remaining results on the PRBench-Core. For each model, we compare the performance of our **PRAgent** against the **Direct Prompt** baseline.

series under two settings: a "thinking" mode that enables Long CoT and a "non-thinking" mode that uses standard inference. As shown in Figure 8 (a), enabling Long CoT did not yield notable gains in average performance. Consistent improvement is observed only on the Engagement metric, and Long CoT even negatively affects other metrics for the Qwen3-235B model. These results indicate that Long CoT is not a universally effective strategy for improving performance on AutoPR tasks.

**Parameter scaling laws also hold in AutoPR scenarios.** In general, increasing a model's parameters is a common way to improve performance. To examine this, we analyze four established LLM series. As shown in Figure 8 (b,c), we observe clear parameter-scaling effects across these models, which is well align with parameter scaling laws (Kaplan et al., 2020; Henighan et al., 2020). While performance generally increases with model size, the trend is not strictly consistent across series. For example, Qwen3-32B can outperform the larger InternVL3-38B, indicating that, for this task, performance does not align uniformly with scale across model families.

**Inference-time scaling does not hold for AutoPR tasks.** To investigate the impact of inference-time scaling, we analyze the relationship between think token count on Qwen3-30B-A3B and the average score on PRBench. Figure 8(d) shows that, contrary to conventional scaling laws, increased inference-time scaling does not yield monotonic performance gains in the AutoPR task. There is no positive trend; instead, we observe a negative correlation between think token count and average score (Pearson's $r = -0.1616$, $p = 0.0003$). We hypothesize that this arises from "specification drift," where excessive, unguided reasoning leads the model to over-interpret instructions, introduce extraneous details, or deviate from core objectives of Fidelity, Alignment, and Engagement.

**In-context Learning does not consistently improve performance.** Our experiments show that In-context Learning (ICL), while always beneficial in other generation tasks (Dong et al., 2022; Qin et al., 2023; 2024; Wang et al., 2024), does not uniformly enhance performance across all metrics. This indicates that the effectiveness of prompting strategies is task- and model-dependent. As demonstrated in Table 7, the impact of ICL varies across models and metrics. For example, Qwen-2.5-VL-7B shows a slight improvement in Fidelity, from 42.42% to 45.01%, but a decrease in Engagement (from 47.91% to 45.79%) and Alignment (from 51.53% to 48.25%). Similarly, Qwen-2.5-VL-72B experiences a drop in Fidelity but an increase in Alignment. These mixed outcomes

| Model Name | Fidelity | | Engagement | | | | | | Alignment | | | | Avg. |
|---|---|---|---|---|---|---|---|---|---|---|---|---|---|
| | A&T Acc. | Factual Score | Hook | Logical Attr. | Visual Attr. | CTA | Prof. Pref. | Broad Pref. | Context Rel. | Vis-Txt Integ. | Hashtag | Plat. Pref. | |
| DeepSeek-R1-Distill-7B[R,T] | 43.27 | 20.39 | 36.53 | 48.30 | - | 18.87 | 40.16 | 45.57 | 33.52 | - | 20.28 | 26.38 | 33.33 |
| + PRAgent | 55.75 | 32.61 | 67.94 | 70.33 | 63.96 | 33.97 | 65.62 | 87.40 | 65.81 | 66.49 | 48.62 | 81.45 | 61.66 |
| Qwen-2.5-VL-7B-Instruct | 48.32 | 36.52 | 61.98 | 46.98 | - | 38.82 | 35.35 | 56.45 | 55.86 | - | 40.72 | 58.01 | 47.90 |
| + PRAgent | 61.75 | 55.69 | 61.76 | 58.66 | 60.24 | 16.06 | 67.97 | 75.00 | 57.43 | 61.64 | 49.65 | 67.09 | 57.74 |
| InternVL3-8B | 51.71 | 44.89 | 70.96 | 53.00 | - | 50.00 | 58.59 | 77.83 | 66.76 | - | 56.28 | 83.98 | 61.40 |
| + PRAgent | 64.08 | 51.06 | 73.47 | 58.49 | 69.90 | 45.85 | 63.28 | 88.77 | 75.49 | 67.33 | 51.44 | 81.93 | 65.92 |
| Qwen3-8B[T] | 51.16 | 42.69 | 73.26 | 52.51 | - | 41.24 | 60.64 | 76.17 | 71.40 | - | 60.61 | 89.65 | 61.93 |
| + PRAgent | 67.95 | 58.96 | 75.00 | 83.53 | 71.97 | 45.30 | 97.56 | 99.22 | 86.86 | 72.74 | 61.50 | 97.95 | 76.54 |
| DeepSeek-R1-Distill-14B[R,T] | 50.67 | 41.73 | 69.47 | 55.39 | - | 30.72 | 57.44 | 71.33 | 64.34 | - | 49.41 | 81.02 | 57.15 |
| + PRAgent | 65.61 | 53.86 | 74.62 | 77.94 | 71.94 | 39.29 | 91.31 | 98.63 | 80.53 | 71.91 | 53.32 | 97.85 | 73.07 |
| InternVL3-14B | 51.63 | 46.51 | 71.06 | 54.17 | - | 54.82 | 53.42 | 76.17 | 68.76 | - | 56.32 | 85.84 | 61.87 |
| + PRAgent | 64.56 | 54.34 | 75.62 | 68.08 | 73.18 | 52.13 | 74.61 | 94.24 | 81.57 | 71.54 | 54.41 | 90.53 | 71.23 |
| Qwen3-14B[T] | 51.12 | 46.33 | 73.73 | 56.45 | - | 39.62 | 68.46 | 80.57 | 72.34 | - | 64.78 | 92.09 | 64.55 |
| + PRAgent | 69.58 | 65.18 | 75.00 | 82.18 | 73.88 | 34.88 | 98.93 | 99.71 | 86.83 | 74.59 | 60.90 | 98.05 | 76.64 |
| GPT-oss-20B[R,T] | 51.71 | 54.89 | 69.97 | 41.63 | - | 44.14 | 71.48 | 72.27 | 71.77 | - | 54.51 | 90.92 | 62.33 |
| + PRAgent | 69.74 | 73.07 | 74.85 | 64.97 | 73.04 | 49.43 | 98.44 | 97.75 | 83.47 | 73.92 | 62.24 | 97.75 | 76.53 |
| Qwen3-30B-A3B[T] | 51.14 | 40.76 | 71.08 | 51.68 | - | 35.63 | 48.44 | 68.46 | 67.43 | - | 60.09 | 81.74 | 57.64 |
| + PRAgent | 69.40 | 54.95 | 74.85 | 80.69 | 72.27 | 30.08 | 96.68 | 98.24 | 85.45 | 73.32 | 65.89 | 97.56 | 74.95 |
| DeepSeek-R1-Distill-32B[R,T] | 50.52 | 41.79 | 69.16 | 57.20 | - | 36.65 | 56.64 | 73.63 | 67.16 | - | 49.63 | 85.16 | 58.75 |
| + PRAgent | 65.85 | 54.88 | 74.10 | 81.44 | 70.65 | 39.53 | 92.86 | 97.26 | 81.25 | 71.58 | 49.12 | 93.64 | 72.68 |
| Qwen-2.5-VL-32B-Instruct | 56.90 | 55.88 | 69.71 | 69.78 | - | 56.20 | 87.01 | 85.84 | 66.18 | - | 52.78 | 88.57 | 68.88 |
| + PRAgent | 71.56 | 69.96 | 74.95 | 82.75 | 75.15 | 53.47 | 98.83 | 99.71 | 83.46 | 75.01 | 61.90 | 97.16 | 78.66 |
| Qwen3-32B[T] | 52.25 | 49.68 | 72.51 | 53.52 | - | 47.97 | 78.22 | 77.93 | 69.60 | - | 61.21 | 90.53 | 65.34 |
| + PRAgent | 71.14 | 64.53 | 75.00 | 83.00 | 74.82 | 42.74 | 98.83 | 99.71 | 86.69 | 75.12 | 60.59 | 98.24 | 77.53 |
| InternVL3-38B | 50.93 | 41.47 | 70.20 | 53.52 | - | 50.07 | 48.05 | 74.22 | 67.44 | - | 51.11 | 82.91 | 58.99 |
| + PRAgent | 66.52 | 53.23 | 74.56 | 72.87 | 74.10 | 48.47 | 84.47 | 96.97 | 83.11 | 73.58 | 50.75 | 96.97 | 72.97 |
| Qwen-2.5-VL-72B-Instruct | 52.78 | 42.61 | 74.10 | 62.51 | - | 57.10 | 56.05 | 82.52 | 74.20 | - | 55.03 | 91.89 | 64.88 |
| + PRAgent | 69.43 | 58.45 | 74.71 | 75.07 | 74.79 | 29.70 | 88.96 | 97.56 | 80.37 | 73.93 | 40.97 | 96.29 | 71.69 |
| GPT-oss-120B[R,T] | 52.64 | 58.45 | 69.34 | 41.79 | - | 41.54 | 74.32 | 72.46 | 72.59 | - | 65.32 | 91.99 | 64.04 |
| + PRAgent | 68.64 | 77.15 | 74.92 | 68.13 | 73.91 | 47.71 | 99.41 | 98.34 | 81.68 | 74.53 | 59.83 | 98.73 | 76.91 |
| Qwen3-235B-A22B[T] | 56.10 | 51.28 | 74.25 | 56.88 | - | 52.20 | 78.03 | 82.81 | 74.49 | - | 68.51 | 95.21 | 68.98 |
| + PRAgent | 67.95 | 66.96 | 75.02 | 83.96 | 74.53 | 44.25 | 98.63 | 99.61 | 87.09 | 75.11 | 60.45 | 98.54 | 77.68 |
| Gemini-2.5-Flash | 54.29 | 43.20 | 74.41 | 62.07 | - | 47.05 | 38.38 | 79.98 | 80.83 | - | 61.47 | 91.80 | 63.35 |
| + PRAgent | 70.43 | 67.97 | 74.53 | 82.88 | 74.41 | 46.61 | 97.46 | 98.73 | 85.32 | 74.64 | 58.30 | 96.97 | 77.28 |
| Gemini-2.5-Pro[R] | 57.05 | 46.46 | 75.29 | 69.70 | - | 45.49 | 46.00 | 86.82 | 81.01 | - | 59.86 | 93.26 | 66.09 |
| + PRAgent | 72.31 | 62.22 | 75.09 | 86.11 | 74.80 | 47.35 | 98.93 | 99.80 | 86.86 | 75.08 | 58.02 | 99.02 | 77.97 |
| GPT-4.1 | 50.98 | 37.77 | 74.80 | 55.53 | - | 42.19 | 48.83 | 77.73 | 73.01 | - | 53.32 | 90.62 | 60.48 |
| + PRAgent | 72.66 | 71.42 | 75.20 | 81.48 | 75.33 | 47.27 | 98.05 | 99.22 | 85.06 | 75.56 | 59.11 | 96.48 | 78.07 |
| GPT-4o | 49.72 | 29.30 | 72.21 | 47.54 | - | 40.97 | 30.86 | 59.77 | 60.15 | - | 52.41 | 54.10 | 49.70 |
| + PRAgent | 66.32 | 45.94 | 75.00 | 75.22 | 74.89 | 49.07 | 77.93 | 98.24 | 81.83 | 74.17 | 52.08 | 97.66 | 72.36 |
| GPT-5[R] | 51.71 | 47.84 | 74.06 | 45.75 | - | 37.68 | 72.75 | 78.81 | 75.00 | - | 50.57 | 94.34 | 62.85 |
| + PRAgent | 67.90 | 72.07 | 75.00 | 80.43 | 75.28 | 34.82 | 98.73 | 99.51 | 86.63 | 75.66 | 52.47 | 98.05 | 76.38 |
| GPT-5-mini[R] | 50.83 | 60.16 | 55.73 | 39.41 | - | 33.30 | 64.55 | 59.08 | 58.70 | - | 39.44 | 79.20 | 54.04 |
| + PRAgent | 71.39 | 82.35 | 74.58 | 68.52 | 73.96 | 42.85 | 99.22 | 98.24 | 82.31 | 73.58 | 52.19 | 95.90 | 76.26 |
| GPT-5-nano[R] | 49.43 | 56.91 | 51.94 | 37.08 | - | 31.43 | 57.13 | 50.29 | 52.65 | - | 51.89 | 71.78 | 51.05 |
| + PRAgent | 71.65 | 73.22 | 73.45 | 60.73 | 70.96 | 35.84 | 96.09 | 93.46 | 74.81 | 68.65 | 56.38 | 91.41 | 72.22 |
| human-authored posts | 53.32 | 47.10 | 45.90 | 42.89 | 70.48 | 30.68 | - | - | 52.34 | 66.34 | 33.92 | - | - |

Table 9: The results on the PRBench. For each model, we compare the performance of our **PRAgent** against the **Direct Prompt** baseline.

suggest that ICL introduces variability that may not uniformly benefit all aspects of the AutoPR task. This calls for further research to identify the conditions under which ICL is most effective.

Overall, our findings highlight the nuanced effects of various strategies on AutoPR performance. While some approaches like parameter scaling show clear benefits, others such as Long CoT reasoning and one-shot prompting yield mixed results. This underscores the importance of tailored strategies that consider the specific demands of the AutoPR task and the characteristics of the models employed.

## K    HUMAN PREFERENCE ANALYSIS ON PRBENCH-CORE

Real-world experiments are characterized by contingency and randomness; prior work (Gopalakrishna Pillai et al., 2025) has thus introduced specialized models for engagement prediction to circumvent potential biases in real-world settings, such as the simultaneous exposure of A/B test posts to the same users. To further validate the performance of our proposed PRAgent, following

| Model | Visual Attr. | Vis-Txt Integ |
|-------|-------------|---------------|
| Qwen-2.5-VL-7B | 74.38 | 68.07 |
| + PRAgent | 60.24 | 61.64 |
| Qwen-2.5-VL-32B | 74.40 | 71.34 |
| + PRAgent | 75.15 | 75.01 |
| Qwen-2.5-VL-72B | 74.20 | 68.67 |
| + PRAgent | 74.79 | 73.93 |
| InternVL3-8B | 74.35 | 69.24 |
| + PRAgent | 69.90 | 67.33 |
| InternVL3-14B | 74.25 | 68.54 |
| + PRAgent | 73.18 | 71.54 |
| InternVL3-38B | 74.41 | 67.58 |
| + PRAgent | 74.10 | 73.58 |

Table 10: Direct comparison of visual metrics between PRAgent's intelligent visual handling and a Naive Visual Baseline. For each model, the first row represents the Naive Visual Baseline, and the highlighted row represents the performance of PRAgent.

| Preference Outcome | Percentage (%) |
|--------------------|----------------|
| PRAgent-Generated Wins | 64.8 |
| Tie | 23.4 |
| Human-Authored Wins | 11.7 |

| Preference Outcome | Percentage (%) |
|--------------------|----------------|
| PRAgent-Generated Wins | 36.7 |
| Tie | 28.1 |
| Human-Authored Wins | 35.2 |

(a) Percentage-based results of the human preference study on PRBench-Core, comparing human-authored posts against PRAgent-generated content (with GPT-5 as the backbone).

(b) Percentage-based results of the human preference study on PRBench-Core, comparing human-authored posts against PRAgent-generated content (with Qwen-2.5-VL-7B-Ins as the backbone).

Goldsack et al. (2022) on science communication, we conducted a human preference study on the **PRBench-Core**. In this study, human annotators were presented with pairs of promotional posts for the same research paper: one generated by **PRAgent** (using GPT-5 as backbone) and the other authored by a human. Annotators were asked to choose which post they preferred based on overall quality, engagement, and clarity, or to declare a tie if they were of comparable quality. The aggregated results are shown in Table 11a, providing a direct measure of our method's performance against human-written content.

Beyond the GPT-5 backbone, we also test PRAgent with a smaller, more accessible model, Qwen-2.5-VL-7B-Ins, on PRBench-Core. As summarized in Table 11b, PRAgent-generated posts are preferred over human-authored ones in only 36.7% of cases (with 28.1% ties), indicating that at current capability levels a 7B-scale model, even with PRAgent, still struggles to match the quality of expert human promotion.

## L    EACH STAGE MATTERS FOR PRAGENT.

To assess the contribution of each stage in PRAgent, we conducted ablations with the Qwen2.5-VL-32B-Ins model, systematically removing or altering core components of the multi-agent pipeline. As shown in Table 12, the results indicate that every specialized stage contributes distinctly to final output quality. (1) First, we bypassed Content Extraction and Structuring (Stage 1). Fidelity declined from 70.76 to 66.38, indicating that the hierarchical summarization helps preserve factual coherence. (2) Second, removing Multi-Agent Content

|  | Fidelity | Engagement | Alignment |
|--|----------|------------|-----------|
| **PRAgent** | 70.76 | 80.81 | 79.38 |
| w/o Stage 1 | 66.38 | 80.89 | 79.79 |
| w/o Stage 2 | 68.75 | 79.59 | 76.29 |
| w/o Stage 3 | 62.94 | 80.10 | 71.36 |

Table 12: Ablation study of PRAgent components using Qwen2.5-VL-32B-Ins.

Synthesis (Stage 2) impaired overall performance, with scores dropping across all three metrics, particularly in Alignment (76.29). (3) Lastly, we removed Platform-Specific Adaptation & Orchestration (Stage 3), which caused the most significant performance drop. This dramatically decreased the

| Method | Avg Time (s) | CPU Util. (Stage 1 / 2+3) | Tokens (In / Out) |
|---|---|---|---|
| PRAgent | 103.7 | 165% / 0.07% | 48,918 / 16,695 |
| Direct Prompt | 66.2 | 0.05% / – | 21,342 / 1,523 |

Table 13: Computational cost per paper on PRBench-Core when using Qwen2.5-VL-32B-Ins as the backbone model.

Alignment score from 79.38 to 71.36 and Fidelity to 62.94, producing a generic-style post instead. In conclusion, these ablations provide strong evidence that each stage of PRAgent is indispensable.

Further, to analyze the effectiveness of PRAgent's intelligent visual handling, we conducted a direct comparison with a Naive Visual Baseline across all six models evaluated on PRBench-Core. In this baseline, each post uniformly uses a screenshot of the corresponding paper's first page as its image. In contrast, PRAgent autonomously selects and prepares what it identifies as the most compelling visual elements from the paper. As shown in Table 10, PRAgent consistently outperforms the Naive Visual Baseline in both Visual Attractiveness and Visual-Textual Integration metrics across all models. This demonstrates that PRAgent's intelligent visual handling significantly enhances the overall quality and engagement of the generated promotional content.

## M    COMPUTATIONAL COST OF PRAGENT

We further profile the computational cost of PRAgent on PRBench-Core using Qwen2.5-VL-32B-Ins as the backbone model. In this setting, the LLM is served on a single A800 GPU, while the PRAgent orchestration code runs on an Apple M2 CPU. For PRAgent, the average wall-clock time per paper is 103.7 seconds. Stage 1 (content extraction and figure/table detection with a CPU-based DocLayout-YOLO model) takes 47.9 seconds and reaches an effective single-core CPU utilization of 165%, whereas Stages 2 and 3 are almost entirely dominated by waiting for LLM API responses, with only 0.07% CPU utilization. Offloading the YOLO model to GPU would substantially reduce Stage-1 latency, and even under a CPU-only setup the end-to-end delay remains moderate.

For the Direct Prompt baseline, the average time per paper is 66.2 seconds, which is also dominated by LLM API latency and uses only 0.05% CPU. In terms of token usage with Qwen2.5-VL-32B-Ins, PRAgent consumes on average 48,918 input tokens and 16,695 output tokens per paper, while the baseline uses 21,342 input and 1,523 output tokens. Table 13 summarizes these measurements. Despite the higher token budget, PRAgent remains a lightweight agentic system that can run comfortably on a single personal machine when using either LLM APIs or a locally hosted model.

## N    LANGUAGE ABLATION ON PRBENCH-CORE

In the main experiments, all RedNote posts are generated in Chinese and X (Twitter) posts are generated in English, matching the dominant language of each platform. To assess whether our evaluation protocol depends on this language choice, we conduct an ablation study on PRBench-Core using Qwen-2.5-VL-32B-Instruct as the backbone model.

In the *language ablation* setting, we swap the languages used for each platform, generating RedNote posts in English and X posts in Chinese, while keeping all other settings unchanged. As shown in Table 14, the scores under the language-swapped setting are nearly identical to those with the default languages for both the Direct Prompt baseline and PRAgent: all deviations across Fidelity, Engagement, Alignment, and the overall average are within 0.2 absolute points. These results indicate that PRBench's LLM-based evaluation is largely insensitive to the surface language of the promotional posts, provided that the underlying content is preserved.

## O    GENERALIZATION TO BIOMEDICAL PROMOTION

To evaluate whether PRBench and PRAgent generalize beyond computer-science-related research, we conduct a pilot study on biomedical promotion. We collect 24 peer-reviewed biomedical papers together with the Twitter posts originally used by their authors to publicize the work, and re-generate promotional posts for the same papers using GPT-5 as the backbone model. For each paper, we compare a direct GPT-5 prompting baseline against the full PRAgent pipeline under the PRBench

| Method | Fidelity | Engagement | Alignment | Overall Avg |
|---|---|---|---|---|
| Direct Prompt | 58.71 | 75.23 | 70.67 | 70.56 |
| + language ablation | 58.77 | 75.25 | 70.64 | 70.57 |
| PRAgent | 72.67 | 80.31 | 79.25 | 78.69 |
| + language ablation | 72.63 | 80.46 | 79.19 | 78.74 |

Table 14: Language ablation on PRBench-Core using Qwen-2.5-VL-32B-Instruct as the backbone. The "+ language ablation" rows correspond to swapping the dominant language of each platform (RedNote: English, X: Chinese) while keeping all other settings fixed.

| Model | Fidelity | | Engagement | | | | | | Alignment | | | | Avg. |
|---|---|---|---|---|---|---|---|---|---|---|---|---|---|
| | A&T Acc. | Factual Score | Hook | Logical Attr. | Visual Attr. | CTA | Prof. Pref. | Broad Pref. | Context Rel. | Vis-Txt Integ. | Hashtag | Plat. Pref. | |
| GPT-5 | 47.66 | 42.29 | 72.92 | 54.07 | - | 18.45 | 68.75 | 75.00 | 72.92 | - | 55.10 | 100.00 | 60.72 |
| + PRAgent | 64.58 | 89.80 | 82.07 | 80.80 | 78.26 | 21.38 | 100.00 | 100.00 | 86.96 | 71.73 | 63.22 | 100.00 | 78.23 |

Table 15: Biomedical-domain results on PRBench-style metrics using GPT-5 as the backbone. We evaluate promotional tweets for 24 biomedical papers using the same metric suite as the main experiments.

evaluation protocol with Qwen-2.5-VL-72B-Ins as the LLM judge. As summarized in Table 15, PRAgent again achieves a decisive advantage: the overall average score improves from 60.72 to 78.23, with large gains in Fidelity and Alignment even though the baseline already uses real author-written tweets as a strong reference.

Unlike computer-science-related fields, biomedical research does not yet have a dense social-media promotion ecosystem, which makes direct engagement metrics unstable. We therefore complement the automatic evaluation with human judgements on the same 24 paper-post pairs. Human annotators are asked to compare the original human-authored tweet with the PRAgent-generated tweet for each paper and decide which one they would prefer to post on their own account, or mark a tie when the quality is comparable. As shown in Table 16, PRAgent is preferred in 79.2% of cases, ties the human tweet in 16.7%, and is judged worse in only 4.2%, indicating strong generalization of PRAgent.

## P  REAL-WORLD STUDY SETTING DETAILS

To validate the practical efficacy of PRAgent, we conducted a 10-day in-the-wild study. The following provides a detailed account of the experimental settings designed to ensure the validity of the results and control for confounding variables.

### P.1  SETUP

Two new, anonymous accounts were created on the social media platform RedNote. To minimize any bias stemming from profile appearance while maintaining a professional look, both accounts were configured with similar styles: For username and profile picture, the accounts were given similar, tech-focused usernames typical of the platform and used stylistically similar avatars to project a consistent identity. For biography, the biography for both accounts was set to "Daily NLP/CV Paper Sharing". This setup ensured that user engagement would be a response to the post content itself, rather than to any perceived identity or branding of the account.

### P.2  PAPER SELECTION CRITERIA.

The study involved 10 recent research papers. These were randomly selected from arXiv preprints submitted in the fields of Natural Language Processing (NLP) and Computer Vision (CV) during August 2025. A key criterion was that these papers had not yet gained significant traction or been promoted by major academic influencers, thereby minimizing the impact of pre-existing public awareness on our engagement metrics.

### P.3  POSTING PROTOCOL.

A strict posting protocol was enforced to ensure a controlled comparison:

| Preference Outcome | Percentage (%) |
|---|---|
| PRAgent-Generated Wins | 79.2 |
| Tie | 16.7 |
| Human-Authored Wins | 4.2 |

Table 16: Human preference study on biomedical-domain promotional tweets. Annotators compare PRAgent-generated tweets (with GPT-5 as backbone) against original human-authored tweets for the same 24 papers.

- **Timing:** Each day, promotional content for the same paper was published by both the PRAgent account (experimental group) and the Direct Prompt account (control group) at the exact same time: 12:00 PM (noon) Beijing Time. This time was chosen to ensure consistency across the experimental period.

- **Frequency:** One paper was promoted per day for 10 consecutive days.

### P.4 CONTENT CONTROL

The core variable was the method of content generation. To create a standardized condition for visual elements, the baseline posts uniformly used a screenshot of the corresponding paper's first page as their image. In contrast, PRAgent autonomously selected and prepared what it identified as the most compelling visual elements from the paper. This allowed us to test PRAgent's entire content creation capability, including both text and visual selection.

### P.5 INTERACTION POLICY.

Throughout the 10-day experimental period, both accounts operated under a strict zero-interaction policy. They did not follow any other users, like or save any external posts, or reply to any comments received on their own posts. This ensured that all recorded engagement metrics (views, likes, saves, etc.) were purely organic and directly attributable to the appeal of the generated content.

## Q SHOWCASE OF GENERATED EXAMPLES

To illustrate the qualitative differences between PRAgent and Direct Prompt, we present several representative examples in Figures 28 to 34. These examples highlight how PRAgent's structured multi-agent approach leads to more engaging, accurate, and platform-tailored promotional content compared to the baseline method.

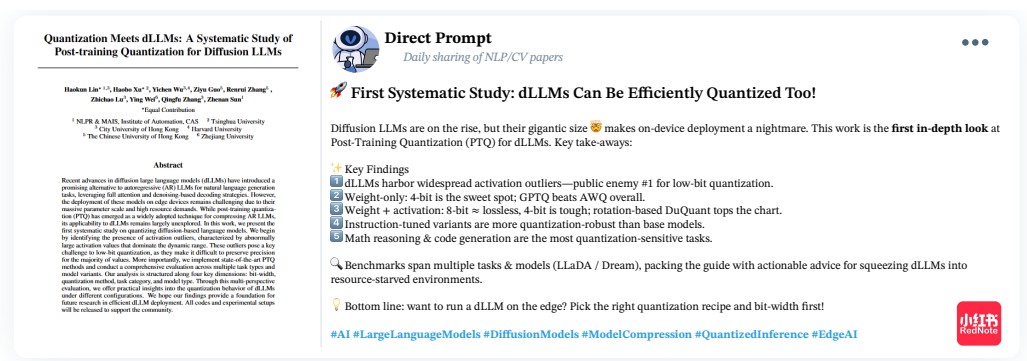

Figure 27: A RedNote post (translated from Chinese to English) generated by Direct Prompt using GPT-5 as the backbone, based on the original paper from Lin et al. (2025).

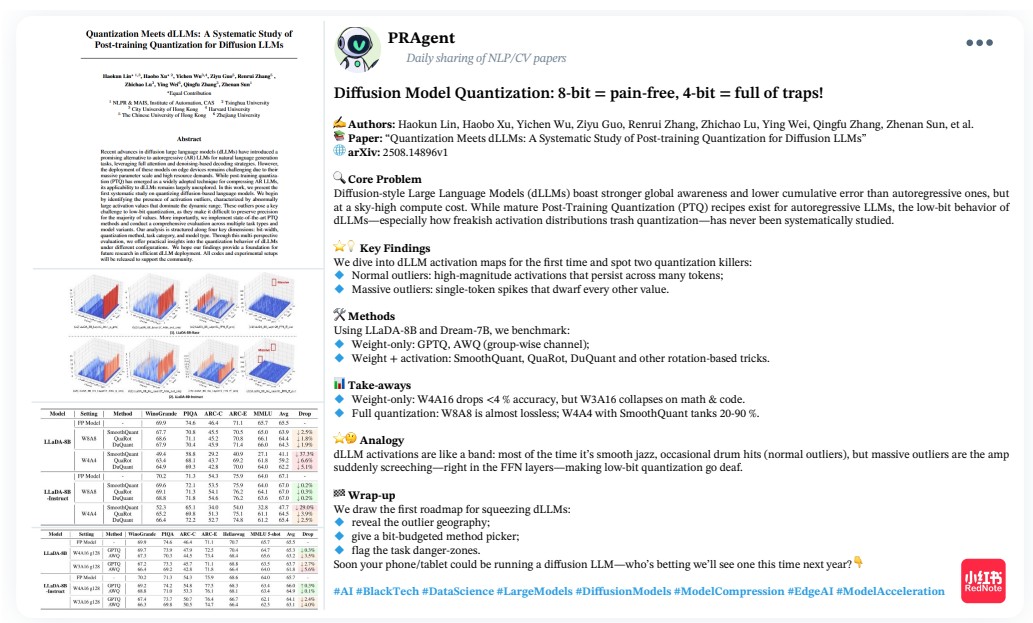

Figure 28: A RedNote post (translated from Chinese to English) generated by PRAgent using GPT-5 as the backbone, based on the original paper from Lin et al. (2025).

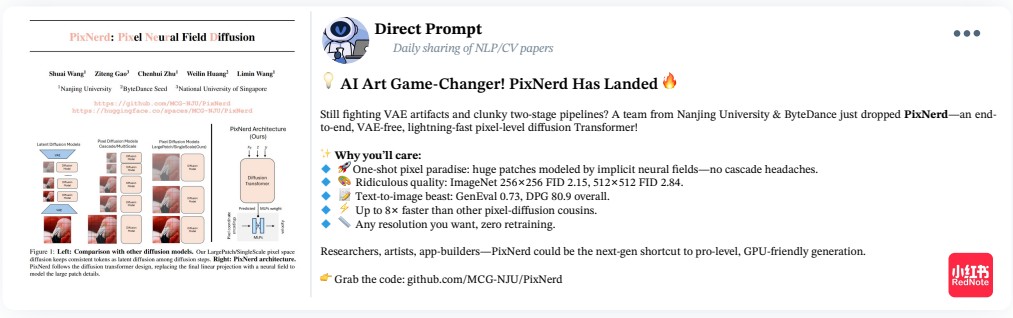

Figure 29: A RedNote post generated by Direct Prompt using GPT-5 as the backbone, based on the original paper from Wang et al. (2025).

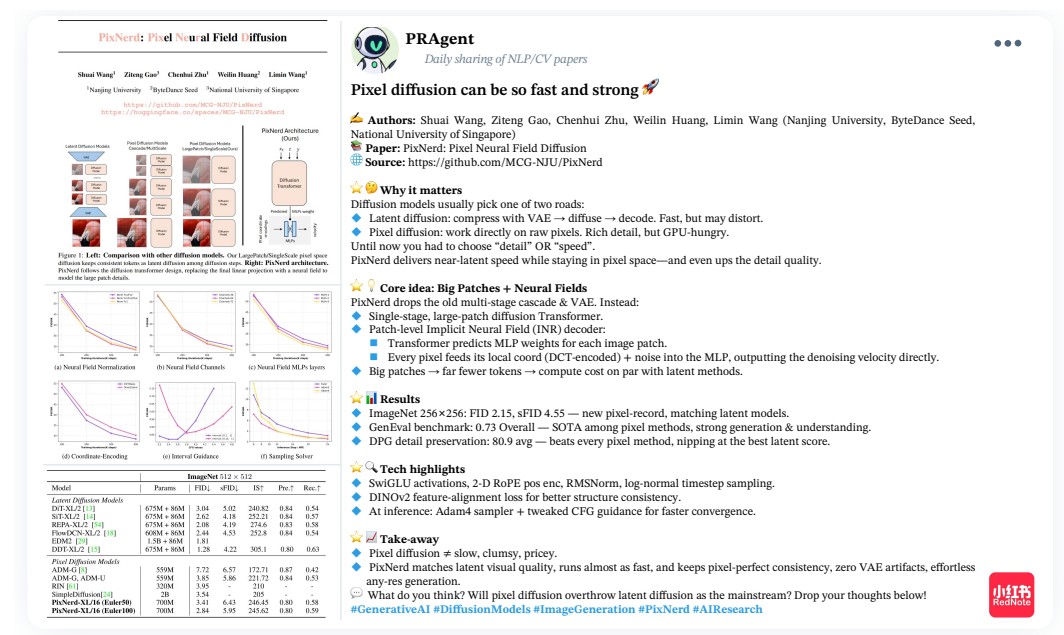

Figure 30: A RedNote post (translated from Chinese to English) generated by PRAgent using GPT-5 as the backbone, based on the original paper from Wang et al. (2025).

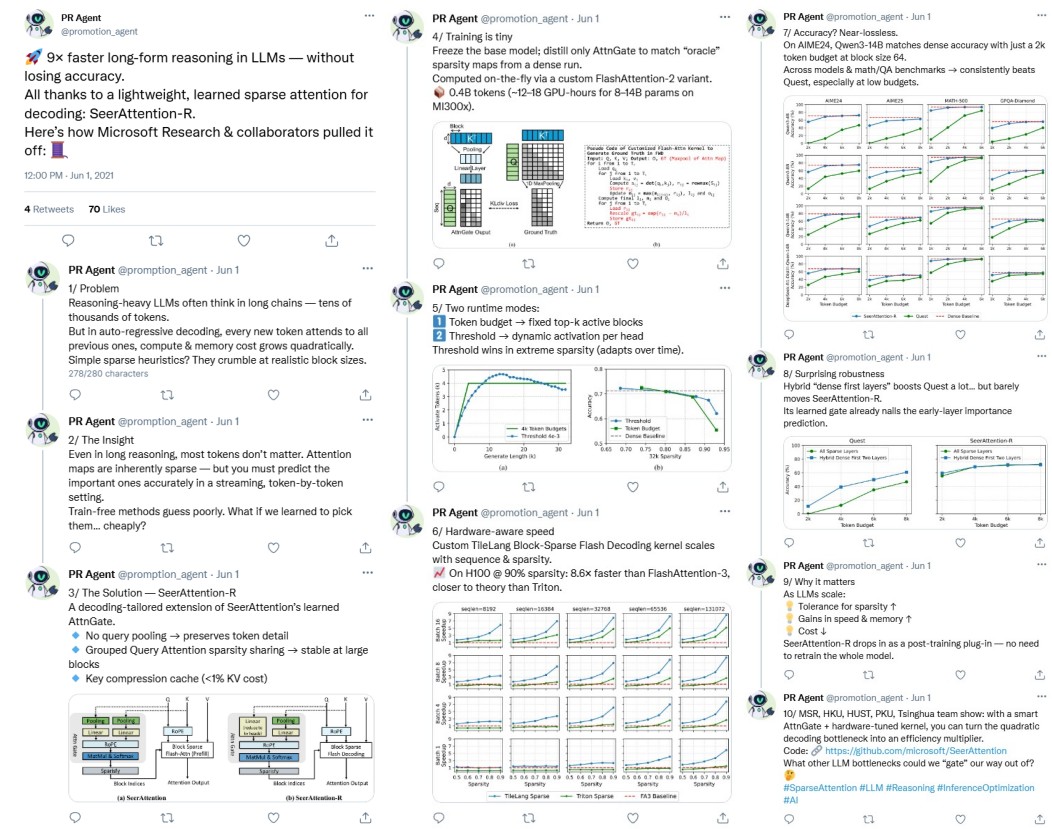

Figure 31: A Twitter post generated by PRAgent using GPT-5 as the backbone, based on the original paper from Gao et al. (2024).

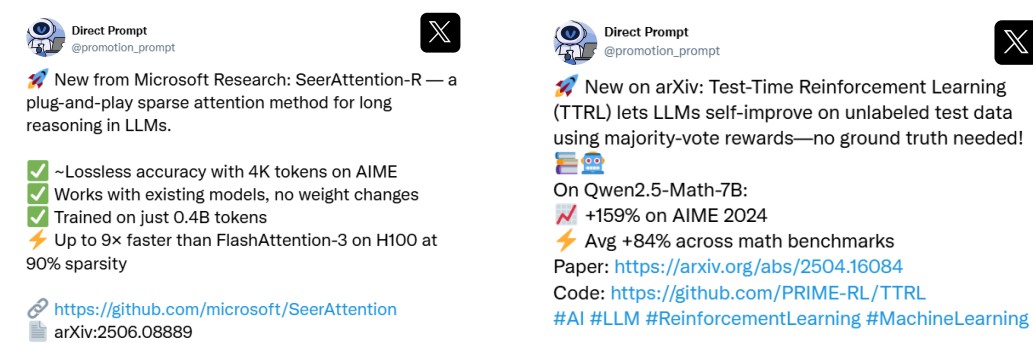

Figure 32: A Twitter post generated by Direct Prompt using GPT-5 as the backbone, based on the original paper from Gao et al. (2024).

Figure 33: A Twitter post generated by Direct Prompt using GPT-5 as the backbone, based on the original paper from Zuo et al. (2025).

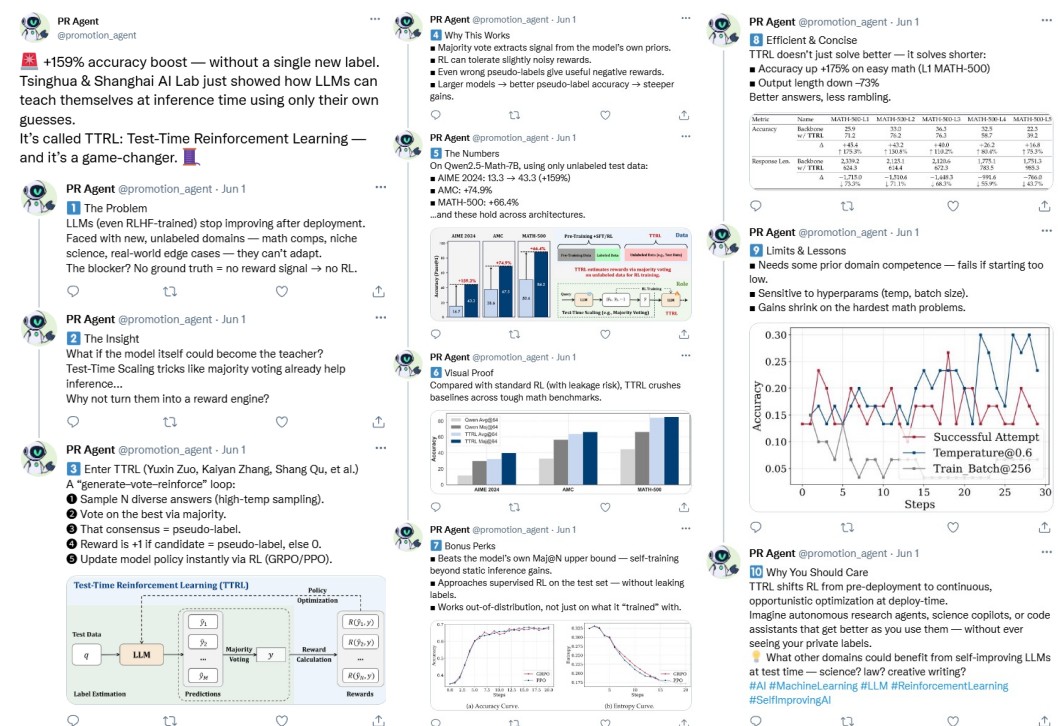

Figure 34: A Twitter post generated by PRAgent using GPT-5 as the backbone, based on the original paper from Zuo et al. (2025).

| Model Series | Version/Sizes (B) |
|---|---|
| GPT-5 Series  (OpenAI, 2025) | nano, mini, chat |
| GPT-4 Series  (Hurst et al., 2024) | 4.1, 4o |
| GPT-OSS Series  (Agarwal et al., 2025) | 20, 120 |
| Gemini 2.5 Series  (Comanici et al., 2025) | Flash, Pro |
| Qwen3 Series  (Yang et al., 2025) | 8, 14, 30, 32, 235 |
| Qwen2.5-VL Series  (Bai et al., 2025) | 7, 32, 72 |
| DeepSeek-R1-Distill Series  (Guo et al., 2025a) | 7, 14, 32 |
| InternVL3 Series  (Zhu et al., 2025) | 8, 14, 38 |

Table 17: All evaluated model list with their versions and sizes.

