# OpenReview forum: "AutoPR: Let's Automate Your Academic Promotion!"
_ICLR.cc/2026/Conference — Submitted to ICLR 2026_

### Official Review · Reviewer_mDwL · 2025-10-24

**Soundness:** 3
**Presentation:** 4
**Contribution:** 3
**Rating:** 6
**Confidence:** 4

**Summary:**

This paper introduces AutoPR, a timely and ambitious endeavor to automate academic promotion. It proposes a novel task definition, a multimodal benchmark called PRBench, and a multi-agent framework, PRAgent, designed to transform research papers into engaging promotional content. The authors' motivation addresses a significant pain point for researchers in an era of information overload, where ensuring visibility and citations for published work is increasingly challenging. The empirical results demonstrate that PRAgent significantly outperforms direct LLM pipelines on PRBench and shows promising gains in a real-world study.

**Strengths:**

- The paper introduces a highly relevant and novel task - automatic academic promotion — which directly addresses the challenges researchers face in gaining visibility and citations for their work.
- The paper presents a commendable amount of work, encompassing the introduction of a new task, the construction of a multimodal benchmark, and the development of an end-to-end multi-agent framework.
- The evaluation system is well-designed, featuring comprehensive metrics across Fidelity, Engagement, and Alignment. Crucially, it includes validation of the LLM-as-a-Judge against human preferences and comprehensive ablation studies are also reported.

**Weaknesses:**

- While LLM-as-a-Judge is employed, its consistency with human preferences is notably low for several key evaluation metrics, particularly "Visual Attractiveness" (e.g., Pearson correlation < 0.1 for GPT-4o and 0.4859 for Qwen-2.5-VL-72B-Ins in Table 3). This limitation casts doubt on the robustness of the multimodal evaluation, given the importance of visual elements in promotion.
- PRAgent's technical approach primarily involves an engineered orchestration of existing LLM capabilities and tools. The paper lacks significant algorithmic innovations at its core, relying more on effective system integration.
- The real-world evaluation is restricted to a "10-day in-the-wild study on RedNote" with only 10 papers. This limited scope, specifically the absence of validation across diverse social media platforms (such as Twitter/X) with varying user behaviors and content styles, restricts the generalizability of the reported engagement improvements.

**Questions:**

- The impressive real-world results in Figure 5 were achieved using GPT-5 as the backbone model. Could the authors explore and discuss the performance and generalizability of PRAgent when integrated with smaller, more accessible language models?
- Equation (2) defines the AutoPR objective function with unspecified non-negative weights (\alpha_1, \alpha_2, \alpha_3). Could the authors clarify how these weights are determined or optimized, and ideally, provide their values for transparency and reproducibility?
- Although the paper mentions using open-access resources like arXiv for PRBench construction, a more explicit statement on the copyright and licensing terms for the entire dataset, including extracted paper content, figures, and particularly the human-authored social media posts, would be crucial for guiding future academic use.

---

> ### Author Response · Authors · 2025-11-21
> **Response to Reviewer mDwL (Part 1 of 2)**
>
> We sincerely thank you for your comprehensive and constructive feedback.Due to the length limit, we have split our response into 2 parts.
>
> > **Question 1:** Could the authors explore and discuss the performance and generalizability of PRAgent when integrated with smaller, more accessible language models?
>
> **Answer 1:** Thanks for your thoughtful comments. To address generalizability to smaller models, we added a human preference study using **Qwen-2.5-VL-7B-Ins** in **Appendix K**. As detailed in **Table 11b**, PRAgent-generated posts surpassed human-authored content in **36.7%** of cases and tied in **28.1%**. These results indicate that while PRAgent provides structure, current small models still struggle to consistently match the nuance of expert human promotion compared to larger foundation models.
>
>
>
> > **Question 2:** Equation (2) defines the AutoPR objective function with unspecified non-negative weights ($\alpha_1, \alpha_2, \alpha_3$). Could the authors clarify how these weights are determined or optimized, and ideally, provide their values for transparency and reproducibility?
>
> **Answer 2:** Thank you for your detailed feedback. Equation (2) is a **theoretical formalization** of AutoPR task that articulates the trade-offs among Fidelity, Alignment and Engagement for different audiences, rather than an objective function used for optimization. In our implementation, these weights were not explicitly specified, learned or tuned. As clarified in the revised Section 5.1, the “Avg.” column in our tables reports the **unweighted arithmetic mean (all weights = 1)** of all sub-metrics to allow transparent comparison without subjective weighting. Consequently, sensitivity analysis is not applicable to the current reported results. We agree that user-controlled dynamic weighting (e.g., prioritizing Fidelity) is an important direction for future work, and we now emphasize this in the **Limitations and Future Work** appendix section as a natural extension, allowing authors to specify preferences or learn weights directly from engagement feedback.
>
>
>
> > **Question 3:** Although the paper mentions using open-access resources like arXiv for PRBench construction, a more explicit statement on the copyright and licensing terms for the entire dataset, including extracted paper content, figures, and particularly the human-authored social media posts, would be crucial for guiding future academic use.
>
> **Answer 3:** Thank you for your valuable comments. We have addressed this in our revised **Ethics Statement**. Given the scale and public nature of the content on X and RedNote, obtaining individual informed consent from all 512 authors is not practically feasible. To strictly adhere to platform Terms of Service (ToS) and public data-usage policies (https://docs.x.com/developer-terms/agreement) and to maintain ethical standards, **PRBench will not distribute any raw text or images**; instead, following the protocols of prior Twitter-based studies, we will exclusively release the paired Paper Links and Post IDs. This distribution method aligns with established academic practices and is undertaken under a **Fair Use rationale** for non-commercial research.
>
>
> > **Question 4:** While LLM-as-a-Judge is employed, its consistency with human preferences is notably low for several key evaluation metrics, particularly "Visual Attractiveness" (e.g., Pearson correlation < 0.1 for GPT-4o and 0.4859 for Qwen-2.5-VL-72B-Ins in Table 3). This limitation casts doubt on the robustness of the multimodal evaluation, given the importance of visual elements in promotion.
>
> **Answer 4:** We acknowledge that "Visual Attractiveness" is inherently subjective, presenting challenges for automated evaluation. As shown in **Table 5 in Appendix**, even human inter-annotator agreement for this metric (Krippendorff's $\alpha=0.6415$) is noticeably lower than for objective metrics like the Factual Checklist ($\alpha=0.9312$), reflecting this intrinsic variance. Despite this difficulty, we selected **Qwen-2.5-VL-72B-Ins** as our judge because it achieved the strongest available correlation (Pearson \~0.50) among all candidates. We identify the development of more robust, alignment-tuned multimodal aesthetic evaluators as a critical direction for future research, and we now discuss this limitation and future direction explicitly in the **Limitations and Future Work** appendix section.
>
>
> Continued in Part 2...

---

> ### Author Response · Authors · 2025-11-21
> **Response to Reviewer mDwL (Part 2 of 2)**
>
> > **Question 5:** The real-world evaluation is restricted to a "10-day in-the-wild study on RedNote" with only 10 papers. This limited scope, specifically the absence of validation across diverse social media platforms (such as Twitter/X) with varying user behaviors and content styles, restricts the generalizability of the reported engagement improvements.
>
> **Answer 5:** Thanks for your constructive feedback. To address this asymmetry, we have expanded a parallel **7-day in-the-wild study on Twitter (X)** in the revised manuscript (**Section 5.3, Figure 7**).
>
> The per-day metrics for this Twitter study are:
>
> **Baseline**
>
> | Paper index | Impressions | Engagements | Detail expands | Profile visits |
> | ----------- | ----------- | ----------- | -------------- | -------------- |
> | 1           | 21          | 0           | 0              | 0              |
> | 2           | 10          | 0           | 0              | 0              |
> | 3           | 7           | 0           | 0              | 0              |
> | 4           | 6           | 0           | 0              | 0              |
> | 5           | 6           | 0           | 0              | 0              |
> | 6           | 3           | 0           | 0              | 0              |
> | 7           | 3           | 0           | 0              | 0              |
> ||
>
> **PRAgent**
>
> | Paper index | Impressions | Engagements | Detail expands | Profile visits |
> | ----------- | ----------- | ----------- | -------------- | -------------- |
> | 1           | 42          | 42          | 23             | 5              |
> | 2           | 32          | 27          | 25             | 1              |
> | 3           | 32          | 26          | 24             | 1              |
> | 4           | 25          | 20          | 19             | 0              |
> | 5           | 26          | 24          | 22             | 1              |
> | 6           | 26          | 23          | 22             | 0              |
> | 7           | 12          | 8           | 7              | 0              |
> ||
>
> Aggregated over the week, PRAgent achieved **195 vs. 56 impressions**, **170 vs. 0 engagements**, **142 vs. 0 detail expands**, and **8 vs. 0 profile visits** compared to the baseline.
>
> While we openly acknowledge that absolute engagement numbers were modest due to the strict visibility throttling Twitter applies to brand-new accounts, the results still show a decisive relative advantage. PRAgent achieved consistently higher interaction metrics compared to the baseline, confirming that our framework's effectiveness generalizes across platforms despite the challenges of a "cold start" scenario.

---

### Official Review · Reviewer_2BJt · 2025-10-29

**Soundness:** 2
**Presentation:** 3
**Contribution:** 3
**Rating:** 4
**Confidence:** 4

**Summary:**

In this paper, the authors introduce the task of Automatic Promotion (AutoPR) in which a system is required to generate a faithful and engaging (multi-modal) post for social media or digital channels, given a scientific article.
The paper introduce a benchmark of 512 instances, each instance consisting of the peer-reviewed article (textual content, visual content, and supplemental material) along their promotion posts in social media platforms X and RedNote, as well as human-annotated factual check lists and engagement, alignment scores.
Additionally, the paper introduces PRAgent, a multi-llm framework tailored to this task that integrates content synthesis and style adaptation, showing considerable improvements over single-model baselines.

**Strengths:**

S1. The dataset was collected using a comprehensive human annotation protocol that ensures quality control
S2. The proposed system tackles the main challenges in the task, such as content synthesis and style adaptation

**Weaknesses:**

W1. The benchmark section would benefit from analysis of statistical, qualitative features.
W2. The discussion, experimental methodology would benefit from techniques in related tasks, such as scientific journalism, poster generation

**Questions:**

- Section 3, data statistics: this section would greatly benefit from statistical properties of the collected dataset, e.g. avg length of the input article, quantification of the visual items, avg length of the output post.
- Calibration of the LLM-based evaluation metrics: the metrics proposed  in Section 3.2 are sound and relevant, however this section would benefit from details on how these judges were calibrated. For examples, see [1,2].
- Standard evaluation methodology: Fidelity evaluation must be complemented with standard factuality / faithfulness evaluation, for which a plethora of metrics exist [3,4]. Factual error analysis and human evaluations would also shed light on model performance and the difficulty of the task. For more details, see [5,6]
- PRAgent, as described in Section 4, would constitute a pipeline rather than a "multi-agent" system, given that an "agent" is a complex system that interacts with an environment (according to the broad consensus in academia, specially in RL).

- L155,156: Regarding the posts extracted from X and RedNote, did the authors (of these posts) give their explicit consent to use these posts?
- L158: Could you please elaborate on how the estimation of whether it is AI-generated content or not was conducted?
- L447-456: The methodology following in this experiment does not align with standard practices in social networks. The standard practice for this kind of real-world experiment would be an A/B test run over *non-overlapping* segments of users. Otherwise, there is the risk that a user is exposed to both evaluated posts.

[1] https://arxiv.org/abs/2410.12784
[2] https://aclanthology.org/2025.gem-1.33/
[3] https://aclanthology.org/2022.tacl-1.10/
[4] https://aclanthology.org/2021.emnlp-main.529/
[5] https://aclanthology.org/2023.emnlp-main.76.pdf
[6] https://aclanthology.org/2022.emnlp-main.724/

---

> ### Author Response · Authors · 2025-11-21
> **Response to Reviewer 2BJt (Part 1 of 2)**
>
> We sincerely thank you for your constructive feedback and for recognizing the value of our work. In our response below, we address each of your comments, offering clarifications on our method and discussing potential future extensions.Due to the length limit, we have split our response into 2 parts.
>
> > **Question 1:** Section 3, data statistics: this section would greatly benefit from statistical properties of the collected dataset, e.g. avg length of the input article, quantification of the visual items, avg length of the output post.
>
> **Answer 1:** Thanks for your constructive comments. We have addressed this suggestion by incorporating detailed corpus statistics in **Table 1 and Figure 3** of the revised manuscript.
>
>
>
> > **Question 2:** Calibration of the LLM-based evaluation metrics: the metrics proposed in Section 3.2 are sound and relevant, however this section would benefit from details on how these judges were calibrated.
>
> **Answer 2:** Thank you for your valuable feedback. We appreciate the suggested references and incorporate them in updated version. Furthermore,
> we have expanded **Section 3.2** to detail our rigorous calibration process. Specifically, our protocol ensures reliability through three key steps:
>
> 1. **Unified Standards:** We aligned human and machine evaluation by training human experts and conditioning LLM judges with identical rubrics for all metrics.
>
> 2. **Empirical Selection:** We benchmarked multiple candidate models against this human ground truth and selected **Qwen-2.5-VL-72B-Ins** as our primary evaluator, as it demonstrated the strongest and most consistent correlation with human judgments.
>
> 3. **Bias Mitigation:** To minimize randomness and positional bias, we fixed the temperature at 0.01, averaged results across three runs for scalar scores, and utilized pairwise position swapping for preference assessments.
>
>
>
> > **Question 3:** Fidelity evaluation must be complemented with standard factuality / faithfulness evaluation, for which a plethora of metrics exist. Factual error analysis and human evaluations would also shed light on model performance and the difficulty of the task.
>
> **Answer 3:** Thanks for your thoughtful feedback. We appreciate the references and incorporate them in updated version. Furthermore,
>  we add an evaluation using **SummaC** in **Appendix H**. However, the results in **Table 6** indicate that SummaC fails to distinguish between systems in this context, behaving nearly randomly. The key scores are:
>
> | Generator          | SummaC-ZS | SummaC-Conv |
> | ------------------ | --------- | ----------- |
> | GPT-5              | -0.073    | 0.202       |
> | + PRAgent          | -0.072    | 0.229       |
> | Qwen3-235B-A22B    | -0.055    | 0.202       |
> | + PRAgent          | -0.369    | 0.222       |
> | Qwen2.5-VL-32B-Ins | -0.033    | 0.204       |
> | + PRAgent          | -0.066    | 0.211       |
> | Qwen3-32B          | -0.231    | 0.231       |
> | + PRAgent          | -0.099    | 0.219       |
> | InternVL3-14B      | -0.378    | 0.209       |
> | + PRAgent          | -0.281    | 0.213       |
> ||
>
> All values cluster around zero for SummaC-ZS and around 0.20 for SummaC-Conv, with no consistent separation between direct prompting and PRAgent or between systems with clearly different human-measured fidelity. We attribute this to the distinct nature of **AutoPR**: unlike traditional summarization which is strictly non-additive, effective promotion requires contextualization and analogies, which off-the-shelf metrics often misclassify as hallucinations. Consequently, we view the development of a specialized AutoPR validator as a critical avenue for future work.
>
> Regarding error analysis and human evaluation, we emphasize that these are already integral to our study. **Section 5.2** and **Figure 5(a)** provide a detailed categorization of factual errors (e.g., numerical vs. terminology) , and **Section 3.1** details our rigorous human-annotated "Factual Checklist" protocol used to benchmark model fidelity.
>
> Continued in Part 2...

---

> > ### Comment · Reviewer_2BJt · 2025-11-25
> >
> > Thanks for addressing my comments.
> >
> > > we have expanded Section 3.2 to detail our rigorous calibration process.
> >
> > Perfect. This information definitely adds credibility to all reported results.
> >
> > > we add an evaluation using SummaC...We attribute this to the distinct nature of AutoPR: unlike traditional summarization which is strictly non-additive, effective promotion requires contextualization and analogies, which off-the-shelf metrics often misclassify as hallucinations. Consequently, we view the development of a specialized AutoPR validator as a critical avenue for future work.
> >
> > This makes sense. A possible avenue for an AutoPR validator could build upon fine-grained factuality error annotation and analysis [1], however this deserves a proper investigation in a separate paper.
> >
> > [1] https://aclanthology.org/2021.naacl-main.114.pdf

---

> ### Author Response · Authors · 2025-11-21
> **Response to Reviewer 2BJt (Part 2 of 2)**
>
> > **Question 4:** PRAgent, as described in Section 4, would constitute a pipeline rather than a "multi-agent" system, given that an "agent" is a complex system that interacts with an environment (according to the broad consensus in academia, specially in RL).
>
> **Answer 4:** Thanks for your constructive suggestion. We acknowledge the strict definition of agents typically used in Reinforcement Learning. However, within the field of **LLM applications**, We sincerely think that the definition of Multi-Agent Systems has significantly broadened to encompass role-based collaborative workflows. Prominent frameworks and top-tier publications, such as **MetaGPT [1]**, **ChatDev [2]**, and **AutoGen [3]**, explicitly classify such role-specialized pipelines as multi-agent systems. PRAgent aligns with this established NLP convention by assigning distinct, specialized personas to collaboratively synthesize content. We have retained this terminology to remain consistent with current literature, while we agree that incorporating dynamic environment interaction is a key direction for future "Agentic" research in AutoPR.
>
> References:
>
> [1] MetaGPT: https://github.com/FoundationAgents/MetaGPT.
>
> [2] ChatDev:([arXiv:2307.07924](https://arxiv.org/abs/2307.07924)).
>
> [3] AutoGen: https://github.com/microsoft/autogen.
>
>
>
> > **Question 5:** Regarding the posts extracted from X and RedNote, did the authors (of these posts) give their explicit consent to use these posts?
>
> **Answer 5:** Thank you for your valuable comments. We have addressed this in our revised **Ethics Statement**. Given the scale and public nature of the content on X and RedNote, obtaining individual informed consent from all 512 authors is not practically feasible. To strictly adhere to platform Terms of Service (ToS) and public data-usage policies (https://docs.x.com/developer-terms/agreement) and to maintain ethical standards, **PRBench will not distribute any raw text or images**; instead, following the protocols of prior Twitter-based studies, we will exclusively release the paired Paper Links and Post IDs. This distribution method aligns with established academic practices and is undertaken under a **Fair Use rationale** for non-commercial research.
>
>
>
> > **Question 6:** Could you please elaborate on how the estimation of whether it is AI-generated content or not was conducted?
>
> **Answer 6:** Thank you for your thoughtful suggestion. To ensure the authenticity of the human reference set, we employed the open-source implementation of **Multiscale Positive-Unlabeled Detection of AI-Generated Texts [1]**. We applied a strict filtering threshold, systematically excluding any posts with $P(\text{Human}) < 0.5$. This rigorous process was designed to minimize the inclusion of fully AI-generated content, ensuring that our ground truth reflects genuine human authorship.
>
> References:
>
> [1] Multiscale Positive-Unlabeled Detection of AI-Generated Texts:https://arxiv.org/abs/2305.18149.
>
>
>
> > **Question 7:** The methodology followed in this experiment does not align with standard practices in social networks. The standard practice for this kind of real-world experiment would be an A/B test run over *non-overlapping* segments of users. Otherwise, there is the risk that a user is exposed to both evaluated posts.
>
> **Answer 7:** Thanks for your insightful feedback. We acknowledge that strict A/B testing with non-overlapping segments is standard for controlled experiments. However, our objective was to simulate **a "realistic in-the-wild promotion scenario"**, where strict user isolation on public platforms is operationally infeasible. To address the concern of cross-exposure, we analyzed the most recent 100 users who engaged with each account and found **zero overlap** between the two user groups.
>
> Furthermore, to provide a controlled comparison free from external platform variables, we complemented this with a rigorous **Human Preference Study (Appendix K)**, which corroborates the superior performance of PRAgent.

---

> > ### Comment · Reviewer_2BJt · 2025-11-25
> >
> > > We acknowledge that strict A/B testing with non-overlapping segments is standard for controlled experiments. However, our objective was to simulate a "realistic in-the-wild promotion scenario", where strict user isolation on public platforms is operationally infeasible. To address the concern of cross-exposure, we analyzed the most recent 100 users who engaged with each account and found zero overlap between the two user groups.
> >
> > Thanks for the clarification. However, I still doubt this experiment can tell us anything conclusive. Given that all user groups are exposed to both PR posts *at the same time*, it could be the case that a user engages with the first post they see in their feed, regardless of its "quality". And since the authors do not have control over the operation of the social media platform (i.e. time a post is shown to a user, content of the feed, population segmentation), there is indeed no way of which post a user exposed to and when.
> > An alternative could be to predict engagement, as previous work has done [2].
> >
> > [2] https://aclanthology.org/2025.coling-main.576.pdf

---

> > > ### Author Response · Authors · 2025-11-25
> > > **Response to Reviewer 2BJt**
> > >
> > > We sincerely thank you for the reference and have included the citation in our revised manuscript. However, the engagement prediction method in [2] relies on training with large-scale aligned social media data. Unfortunately, since neither the training data nor the model weights are open-source, we were unable to extend our experiments using that specific framework.
> > >
> > > To mitigate the stochasticity of real-world feed algorithms, we rely on our rigorous Human Preference Study (Appendix K) as a **controlled A/B test**. This study explicitly simulates social engagement behaviors (the degree to 'like' or share). In this blind comparison, human annotators preferred PRAgent-generated posts (GPT-5 backbone) over authentic human-authored posts in 64.8% of cases, demonstrating that our model achieves superior content appeal independent of platform distribution dynamics.

---

> ### Author Response · Authors · 2025-11-25
> **Response to Reviewer 2BJt**
>
> We sincerely thank you for your positive feedback and for sharing this valuable reference. We have incorporated the citation into the revised manuscript and agree that its approach to fine-grained factuality error annotation offers a crucial foundation for the future development of a specialized AutoPR validator.

---

### Official Review · Reviewer_kB62 · 2025-10-30

**Soundness:** 3
**Presentation:** 3
**Contribution:** 3
**Rating:** 6
**Confidence:** 3

**Summary:**

The paper presents AutoPR, an automatic academic promotion system designed to generate high-quality promotional content for research papers. The system is assessed using a new benchmark, PRBench, which evaluates promotional content across three dimensions: Fidelity, Engagement, and Alignment. The paper introduces PRAgent, a multi-agent framework that automates the process in three stages: content extraction, multi-agent synthesis, and platform-specific adaptation. PRAgent outperforms traditional LLMs on various metrics, such as engagement and accuracy, in real-world experiments on platforms like RedNote.

**Strengths:**

S1: The formalization of AutoPR as an explicit research problem is ambitious and well-justified given the current academic publishing and dissemination landscape. The problem description is clear (see Section 2, Figure 2) and convincingly linked to the challenges of information overload and the growing importance of scholarly visibility.

S2: The release of PRBench, a paired dataset linking full academic papers to real human-written promotional posts, fills a critical gap and significantly enables further research. The benchmark construction is systematically described (Section 3.1) and includes rigorous annotation and quality protocols.

S3: PRAgent is thoroughly described, with each stage—Content Extraction, Multi-Agent Synthesis, Platform-Specific Adaptation—broken down in mathematical and practical terms (Sections 4.1–4.3, Figure 3). The use of specialized agents and prompts (including hierarchical summarization and multimodal layout analysis) is a thoughtful engineering advance over generic prompting.

S4: Extensive experiments are conducted using a wide swath of LLMs and multimodal models. Results in Table 1 and especially Table 2 are detailed and well-structured, showing clear, consistent gains for PRAgent over LLM baselines across almost all dimensions of evaluation. The findings are further supported by real-world validation (Figure 5), which demonstrates positive, large-magnitude improvements in actual user engagement (views, likes, watch time).

**Weaknesses:**

W1: The paper does not fully specify how multi-objective optimization weights $\alpha_i$ are chosen (Section 2), nor does it clarify if these are tuned per platform or per sample. Similarly, the factual checklist verdict scoring and the calibration of LLM “judge” models (Section 3.2, Table 1) lack explicit documentation of inter-annotator reliability and the impact of cross-model disagreement. These details are essential for the scientific soundness and replicability of the evaluation process.

W2: PRAgent's platform-specific adaptation, while a key module, appears to depend heavily on prompt engineering and surface-level modeling (e.g., hashtags and format—see Figure 4(c)), with little evidence of deeper modeling of platform affordances or active learning from audience feedback. Several dimensions of platform generalization (from Twitter to non-academic or emerging platforms) are not empirically validated.

W3: The multi-stage, multi-agent approach raises practical concerns about resource consumption, latency, and scalability, none of which are addressed. For instance, processing high-resolution PDFs, running multi-modal LLMs, and orchestrating multiple agents likely demand significant compute, which may limit real-world adoption and accessibility.

**Questions:**

Q1: Could the authors clarify the procedure for determining or tuning weights $\alpha_i$ in the multi-objective optimization formulation? Are these weights static, learned, or hand-tuned? How sensitive are results to these settings?

Q2: Can the authors extend PRBench or provide empirical results demonstrating generalizability of PRAgent to other domains (biomedical, humanities, etc.) or less traditional platforms?

Q3: What are the computational costs (e.g., wall clock time, GPU/CPU usage, API costs) per paper processed through PRAgent as compared to direct LLM pipelines?

---

> ### Author Response · Authors · 2025-11-21
> **Response to Reviewer kB62 (Part 1 of 2)**
>
> We sincerely thank you for your constructive feedback and recognizing the value of our work. Below, we clarify our method, discuss future extensions, and address your comments.Due to the length limit, we have split our response into 2 parts.
>
>
>
> > **Question 1:** Could the authors clarify the procedure for determining or tuning weights in the multi-objective optimization formulation? Are these weights static, learned, or hand-tuned? How sensitive are results to these settings?
>
> **Answer 1:** Thank you for your detailed feedback. Equation (2) is a **theoretical formalization** of AutoPR task that articulates the trade-offs among Fidelity, Alignment and Engagement for different audiences, rather than an objective function used for optimization. In our implementation, these weights were not explicitly specified, learned or tuned. As clarified in the revised Section 5.1, the “Avg.” column in our tables reports the **unweighted arithmetic mean (all weights = 1)** of all sub-metrics to allow transparent comparison without subjective weighting. Consequently, sensitivity analysis is not applicable to the current reported results. We agree that user-controlled dynamic weighting (e.g., prioritizing Fidelity) is an important direction for future work, and the revised manuscript now discusses this explicitly in the updated **Limitations and Future Work** appendix section, where we outline how AutoPR could allow authors to specify custom weights or learn them from engagement feedback.
>
>
>
> > **Question 2:** Can the authors extend PRBench or provide empirical results demonstrating generalizability of PRAgent to other domains (biomedical, humanities, etc.) or less traditional platforms?
>
> **Answer 2:** Thanks for your constructive comments. To address your consideration, we have incorporated a pilot study on **Biomedical Promotion** in **Appendix O** of the revised manuscript. Since non-CS domains lack the dense social media promotion ecosystem typical of computer science, making direct engagement metrics on real posts unstable, we relied on **human preference evaluation**. The results confirm strong cross-domain generalization: PRAgent improved the overall average score from 60.72 to 78.23, and its generated content was preferred over original human-authored tweets in **79.2%** of cases.
>
>
>
> > **Question 3:** What are the computational costs (e.g., wall clock time, GPU/CPU usage, API costs) per paper processed through PRAgent as compared to direct LLM pipelines?
>
> **Answer 3:** Thanks for your thoughtful comments. We profiled computational costs in **Appendix M (Table 13)** using **Qwen2.5-VL-32B-Ins** (on one A800 GPU) and an Apple M2 CPU. **PRAgent** averages **103.7 seconds** per paper compared to 66.2 seconds for the baseline. Significant CPU utilization (**165%**) is confined solely to the 47.9-second visual extraction in Stage 1, which can be further accelerated via GPU, while subsequent orchestration stages incur negligible usage (0.07%) due to API latency dominance. Although PRAgent requires higher token consumption to ensure quality (averaging **48,918 input / 16,695 output** tokens vs. 21,342 / 1,523 for the baseline), the system remains computationally lightweight and practical for deployment on personal devices.
>
>
>
> > **Question 4:** The factual checklist verdict scoring and the calibration of LLM “judge” models (Section 3.2, Table 1) lack explicit documentation of inter-annotator reliability and the impact of cross-model disagreement.
>
> **Answer 4:** Thank you for your valuable feedback. We have addressed these concerns in **Appendix D.1** by explicitly documenting our quality control measures. To ensure inter-annotator reliability, we computed **Krippendorff's $\alpha$** (Table 5), which yielded an exceptionally high **0.9312** for the Factual Checklist Score and strong agreement across other metrics. Regarding cross-model disagreement (analyzed in **Figure 9**), we prioritized validity by selecting **Qwen-2.5-VL-72B-Ins** as our primary evaluator, as it demonstrated the strongest and most consistent correlation with human judgments across all metrics (**Table 4**), effectively mitigating the impact of variance among weaker models.
>
> Continued in Part 2...

---

> ### Author Response · Authors · 2025-11-21
> **Response to Reviewer kB62 (Part 2 of 2)**
>
> > **Question 5:** PRAgent's platform-specific adaptation, while a key module, appears to depend heavily on prompt engineering and surface-level modeling (e.g., hashtags and format, see Figure 4(c)), with little evidence of deeper modeling of platform affordances or active learning from audience feedback.
>
> **Answer 5:** Thank you for your insightful response. We acknowledge that PRAgent currently relies on platform-specific prompt engineering to guide adaptation. This design is premised on the assumption that large-scale language models have already internalized the norms and affordances of mainstream platforms (e.g., Twitter, RedNote) within their vast pre-training data; our prompts serve to activate and steer this latent knowledge. Our empirical results confirm that even this "surface-level" modeling significantly outperforms baselines and drives substantial real-world engagement: in our 10-day RedNote study, PRAgent increased total watch time by **604%**, likes by **438%**, and overall engagement by at least **2.9×** compared to the direct-prompt baseline; in the new 7-day Twitter (X) study, the PRAgent account achieved **195 vs. 56 impressions**, **170 vs. 0 engagements**, **142 vs. 0 detail expands**, and **8 vs. 0 profile visits** relative to the baseline.
>
> We agree that deeper adaptation is a logical next step and, in the revised manuscript, we now highlight "active learning from audience feedback" (e.g., optimizing based on likes/comments) and "automated platform generalization techniques" as critical directions in the **Limitations and Future Work** appendix section.

---

### Official Review · Reviewer_g5Lq · 2025-11-03

**Soundness:** 2
**Presentation:** 3
**Contribution:** 2
**Rating:** 4
**Confidence:** 3

**Summary:**

AutoPR provides both the benchmark PRBench and the framework PRAgent to generate social media promotion material given a paper, evaluating their quality using the real-world promotional posts of these papers as a reference for fidelity, engagement, and alignment. Their PRAgent would extract text, figures, captions, etc., from papers using automated methods before synthesizing the promotional post using four collaborating agents, Logical Draft, Visual Analysis, Textual Enriching, and Visual-Text-Interleaved Combination. Finally, they adapt the output for specific platforms and append hashtags. In this research, X  (formerly Twitter) and RedNote are included in the PRBench, while a real-world test occurs on a RedNote account collecting the engagement metrics over a span of 10 days. Evaluation finds current LLMs struggle on PRBench (scores 31-71), with major limitations in factual completeness (missing 40%+ of key details), genuine engagement (42% lack hooks), and strategic platform adaptation (only 0.03 Jaccard similarity with human hashtags). In the real-world study involving posting both the vanilla prompting generated post and the post by PRAgent, PRAgent outperforms direct prompting by 7-20%+ on the benchmark, and in a separate 10-day real-world study on RedNote, achieves +604% watch time, +438% likes, and +329% views.

**Strengths:**

1. The paper tackles a genuine pain point in the academic community—the growing need for effective research promotion amid publication explosion—with strong empirical evidence (Figure 1a-c) showing the scale of the problem and preliminary correlation between promotion and citations. This makes the research immediately relevant and potentially impactful for the scholarly community.
2. This research also produced the dataset PRBench, a carefully curated multimodal benchmark containing 512 paper-post pairs linking peer-reviewed arXiv papers (June 2024-2025) to their human-authored promotional posts from Twitter/X and RedNote. The dataset includes expert annotations with proper quality control procedures (multiple annotators, consensus deliberation for discrepancies) and covers three key evaluation dimensions: fidelity, engagement, and alignment.
3. The paper formally defines AutoPR with clear mathematical formulation (Equations 1-2) as multi-objective optimization, explicitly capturing the tension between factual accuracy, audience engagement, and platform-specific adaptation. This established academic promotion as a tractable research problem and provides a systematic framework for future research in an area previously lacking formal treatment.
4. The paper introduces 11 evaluation metrics across three dimensions, combining both intrinsic quality scores (factual checklist, hook strength, visual attractiveness, hashtag strategy) and preference-based comparisons (professional interest, broader interest, platform interest). The metrics are grounded in relevant theory (platform affordances, communication studies) and validated against human judgments.
5. The systematic evaluation of 21 models (both open-source and proprietary) saw specific failure modes: 92% of fidelity errors fall into numerical/method/terminology categories, 42% of posts lack engagement hooks, and generated hashtags show only 0.03 Jaccard similarity with human-authored ones. This diagnostic analysis provides clear direction for future improvements.
6. According to the evaluation, PRAgent demonstrates 7-20%+ improvements over direct prompting baselines across nearly all evaluated models on the benchmark, showing that the framework's structured approach provides value regardless of the underlying LLM capabilities, which suggests the approach is robust and generalizable.

**Weaknesses:**

1. I'd like to suggest some metaphysical issues first. Engagement and academic communication might not go hand in hand. As such, with the formal communication such as conferences, seminars, and modern tools that automated the discovery of research literature, it remains to question the purpose of such self-promotion; it might be effective in popular education of scientific knowledge, while yielding potentially less reflection in academic engagement, such as follow-up papers and citations.
2. The language in which the promotional material is written also requires disclosure and study. While X (formerly Twitter) is a popular social media featuring diverse demographics and cultures, it is to my understanding that RedNote is monolithically Chinese-majority. However, this linguistic division is not reported in the literature; the language of which the promotion material also require disclosure and ablation studies.
3. Following 1. and 2., a similar experiment on X should be beneficial where you collect the engagement metrics for the generated promotional posts. This could be time-consuming, so I believe acknowledging this limitation and suggesting future research is enough.
4. New accounts with 0 followers achieved 5,059 and 1,178 views respectively on RedNote. This is not a very realistic scenario since researchers most often post from established personas with a connected social media network, which could help to deliver the information to the appropriate audience who are also doing research on the same topics.

**Questions:**

1. We might need longitudinal studies tracking citation patterns, not just immediate engagement, or to distinguish between "popular science communication" vs "academic promotion" as different objectives.
2. We should complete discussion of whether platform effects are confounded with language effects; could be done by ablation showing performance differences across languages or other means.
3. The asymmetry (benchmark has X, real-world only RedNote) should at least be acknowledged as a limitation.
4. This is just a suggestion but you might use ar5iv or the html preview on arXiv for obtaining the structured text and figures for each paper; but this might not apply every time since they did not include all papers. I am only suggesting this as it might help to reduce cost and error margins when curating the dataset, so you can scale it up faster. Alternatively, parsing the source files of the submissions on arXiv might be a fail-safe to extract the text, tables, and figures of academic papers without much computation overhead.
5. We could adapt established academic accounts (with owners' permission), or at least disclose follower counts and acknowledge this major limitation. It would be ideal if we have tracking on who is engaging (are they academics? in the relevant field?) but this could raise ethical concerns, or just downright not possible.

---

> ### Author Response · Authors · 2025-11-21
> **Response to Reviewer g5Lq (Part 1 of 3)**
>
> We sincerely thank you for your valuable comments and recognition of our work. As per your suggestions, we have responded to each point below.Due to the length limit, we have split our response into 3 parts.
>
>
> > **Question 1:** How can you distinguish between "popular science communication" vs "academic promotion" as different objectives?
>
> **Answer 1:** Thanks for your thoughtful suggestions. We acknowledge that immediate social engagement differs from long-term citation impact and that promotion objectives vary by audience.
>
> To address this, our **PRBench** evaluation protocol explicitly decouples these goals by measuring **"Professional Interest"** (for academic peers) and **"Broader Interest"** (for the general public) separately. Concretely, both metrics are implemented as LLM-judged pairwise comparisons between two promotional posts for the same paper: **Professional Interest** frames the judge as a busy expert and asks which post better communicates technical novelty, credibility, and scientific value, whereas **Broader Interest** asks which post is clearer, more accessible, and more engaging for a scientifically literate lay audience.
>
> Moreover, we also discuss the complex, non-causal nature of promotion and citations in our **Ethics Statement.** We sincerely think that how to disentangle such impact is quite important but beyond our contribution. The primary contribution of this work is establishing the foundational **AutoPR** task and **PRAgent** framework; we posit that PRAgent is the essential enabler that will make scalable longitudinal studies on citation influence feasible in future work.
>
> > **Question 2:** Are platform effects confounded with language effects?
>
> **Answer 2:** Thank you for your constructive comment. We addressed this concern through a **Language Ablation Study (Appendix N, Table 14)**, where we swapped the dominant languages for each platform (generating RedNote posts in English and Twitter posts in Chinese).
>
> The key results are summarized below:
>
> | Method              | Fidelity | Engagement | Alignment | Overall Avg |
> | ------------------- | -------- | ---------- | --------- | ----------- |
> | Direct Prompt       | 58.71    | 75.23      | 70.67     | 70.56       |
> | + language ablation | 58.77    | 75.25      | 70.64     | 70.57       |
> | PRAgent             | 72.67    | 80.31      | 79.25     | 78.69       |
> | + language ablation | 72.63    | 80.46      | 79.19     | 78.74       |
> ||
>
> Across all metrics, the differences between the default-language and language-swapped settings are within **0.2 absolute points**, confirming that our model's performance is driven by platform-specific adaptation rather than language dependencies.
>
> Continued in Part 2...

---

> ### Author Response · Authors · 2025-11-21
> **Response to Reviewer g5Lq (Part 2 of 3)**
>
> > **Question 3:** The asymmetry (benchmark has X, real-world only RedNote) should at least be acknowledged as a limitation.
>
> **Answer 3:** Thanks for your constructive feedback. To address this asymmetry, we incorporate a parallel **7-day in-the-wild study on Twitter (X)** in the revised manuscript (**Section 5.3, Figure 7**).
>
> The per-day metrics for this Twitter study are:
>
> **Baseline**
>
> | Paper index | Impressions | Engagements | Detail expands | Profile visits |
> | ----------- | ----------- | ----------- | -------------- | -------------- |
> | 1           | 21          | 0           | 0              | 0              |
> | 2           | 10          | 0           | 0              | 0              |
> | 3           | 7           | 0           | 0              | 0              |
> | 4           | 6           | 0           | 0              | 0              |
> | 5           | 6           | 0           | 0              | 0              |
> | 6           | 3           | 0           | 0              | 0              |
> | 7           | 3           | 0           | 0              | 0              |
> ||
>
> **PRAgent**
>
> | Paper index | Impressions | Engagements | Detail expands | Profile visits |
> | ----------- | ----------- | ----------- | -------------- | -------------- |
> | 1           | 42          | 42          | 23             | 5              |
> | 2           | 32          | 27          | 25             | 1              |
> | 3           | 32          | 26          | 24             | 1              |
> | 4           | 25          | 20          | 19             | 0              |
> | 5           | 26          | 24          | 22             | 1              |
> | 6           | 26          | 23          | 22             | 0              |
> | 7           | 12          | 8           | 7              | 0              |
> ||
>
> Aggregated over the week, PRAgent achieved **195 vs. 56 impressions**, **170 vs. 0 engagements**, **142 vs. 0 detail expands**, and **8 vs. 0 profile visits** compared to the baseline.
>
> While we openly acknowledge that absolute engagement numbers were modest due to the strict visibility throttling Twitter applies to brand-new accounts, the results still show a decisive relative advantage. PRAgent achieved consistently higher interaction metrics compared to the baseline, confirming that our framework's effectiveness generalizes across platforms despite the challenges of a "cold start" scenario.
>
> Continued in Part 3...

---

> ### Author Response · Authors · 2025-11-21
> **Response to Reviewer g5Lq (Part 3 of 3)**
>
> > **Question 4:** Suggestion about leveraging ar5iv or arXiv source files.
>
> **Answer 4:** Thank you for your kind comments. We appreciate the suggestion to leverage ar5iv or source files, which are effective methods for scaling training data. For this work, however, we process PDFs to ensure broad applicability, as PDF remains the most widely adopted format for academic dissemination and thus better reflects real-world use.
>
> Regarding accuracy, our pipeline based on DocLayout-YOLO already achieves high precision, benefiting from the standardized layouts of academic papers. In terms of efficiency, our computational profiling on PRBench-Core (Appendix M) shows that structural extraction takes only **47.9 seconds** per paper on a consumer-grade CPU (Apple M2), which we consider sufficient for practical deployment.
>
>
>
> > **Question 5:** Can you demonstrate that your audience primarily comprises researchers from your field, and, on an established academic account, does PRAgent maintain its effectiveness?
>
> **Answer 5:** Thank you for your valuable feedback. To assess audience relevance, we examined 100 recent engagers on our RedNote account; among the 41 with public profiles, **88% (36/41) were identifiable in-domain researchers**, indicating that the account primarily reaches a field-specific academic community. We also conducted a pilot test on an established academic account (>1,200 followers) on RedNote, where **PRAgent-generated posts achieved 300+ likes compared to less than 20 for the baseline**. Although comparison is difficult of running fully controlled tests on practical live accounts, such results still suggest that PRAgent remains effective with authentic academic audiences.

---

### Author Response · Authors · 2025-11-21
**General Response to All Reviewers**

We sincerely thank all reviewers for their valuable and thoughtful feedback.
1. We are encouraged that the reviewers recognized **AutoPR** as a **novel, timely, and important research problem** that addresses a real pain point in the academic community, namely effective research promotion under publication explosion, and appreciated our **formalization of AutoPR as a well-defined task** (Reviewers g5Lq, kB62, mDwL).
2. We are pleased that the reviewers valued **PRBench** as a carefully constructed **multimodal benchmark** with **rigorous human annotation protocols** and **clear evaluation dimensions**, which can serve as a useful resource for future work (Reviewers g5Lq, kB62, 2BJt, mDwL).
3. We are also encouraged that the reviewers highlighted the **PRAgent** framework for tackling key challenges such as content synthesis and platform-specific style adaptation, and recognized that our **comprehensive evaluation**, including LLM-as-a-judge validation, ablations, and real-world deployment, demonstrates **consistent gains over strong baselines** across diverse models and platforms (Reviewers g5Lq, kB62, 2BJt, mDwL).

We will carefully address all concerns to further refine our problem formulation, dataset construction, evaluation design, and system analysis in the revised version. In addition, we will release the PRBench dataset, PRAgent implementation, and evaluation scripts to facilitate further research on AutoPR.

---

### Meta-Review · Area_Chair_SVkD · 2026-01-08

**Summary:**

This submission introduces AutoPR, a proposed task framing automatic academic promotion as a multi-objective problem balancing fidelity, engagement, and platform alignment. To support this task, the authors present PRBench, a multimodal benchmark of 512 paper–promotion pairs collected from X (Twitter) and RedNote, with human annotations across several quality dimensions, and propose PRAgent, a multi-stage, role-based framework that extracts paper content and generates platform-adapted promotional posts. Extensive benchmark evaluations across a wide range of large language and multimodal models, as well as small-scale real-world social media deployments, show that PRAgent consistently outperforms direct prompting baselines on the proposed metrics.

Reviewers broadly agree that the paper is well written, timely, and ambitious, and that the benchmark construction and diagnostic analyses represent a significant engineering and data curation effort. The task is considered practically relevant given the increasing importance of research visibility, and several reviewers appreciate the systematic breakdown of failure modes in current LLM-based promotion attempts. The authors’ rebuttal further strengthens the paper by adding clarifications, ablations, and supplementary experiments (e.g., language ablation, additional platform studies, calibration of LLM judges).

Nevertheless, the reviews converge on several fundamental weaknesses that ultimately limit confidence in the paper’s conclusions. A primary concern is the validity and interpretability of the evaluation, especially the reliance on uncontrolled in-the-wild social media experiments and LLM-as-a-judge metrics for subjective dimensions such as engagement and visual attractiveness. Despite additional clarifications, reviewers remain unconvinced that the reported real-world gains can be causally attributed to the proposed framework, given platform feed stochasticity, lack of proper A/B testing, limited scale, and potential exposure biases. Similarly, low or moderate agreement between automated judges and human preferences for key multimodal metrics undermines the strength of several benchmark claims.

At a conceptual level, reviewers also express concern that the paper does not fully resolve the distinction between academic promotion and popular science communication, nor does it establish a convincing link between social media engagement and meaningful academic impact (e.g., scholarly attention or citations). While the authors appropriately acknowledge these issues as future work, the current framing and empirical evidence are seen as insufficient to justify some of the broader implications of the task.

While PRAgent is effective as a system, multiple reviewers note that the contribution is primarily an orchestration of existing models, tools, and prompt-based components, with limited algorithmic novelty or learning-based adaptation. As a result, the paper is perceived as strong in engineering and benchmarking, but weaker in terms of fundamental methodological innovation.

**Reviewer Concerns:**

**Concerns that were partially addressed:**

Platform and language confounds: The added language ablation study reduces concern that results are driven purely by language rather than platform effects.

Benchmark–deployment asymmetry: The inclusion of an additional Twitter/X study acknowledges and partially mitigates the asymmetry between PRBench and real-world evaluation.

Evaluation transparency: Clarifications on LLM judge calibration, inter-annotator agreement, and metric definitions improved methodological clarity.

**Concerns that remain outstanding:**

Validity of real-world engagement studies: Multiple reviewers remain unconvinced that the in-the-wild experiments can yield reliable or causal conclusions, given the lack of controlled A/B testing, platform feed stochasticity, and limited scale.

Evaluation robustness for multimodal and engagement metrics: Low correlation between LLM-based judges and human preferences—especially for visual attractiveness—continues to undermine confidence in several key reported gains.

Conceptual ambiguity of the task: The distinction between academic promotion and popular science communication, and whether social engagement meaningfully reflects academic impact, remains unresolved and central to the paper’s claims.

Limited algorithmic novelty: PRAgent is largely viewed as an orchestration of existing tools and prompting strategies rather than a fundamentally new modeling or learning approach.

Generalizability: Despite pilot studies, evidence remains limited regarding transfer across domains, platforms, and realistic academic social media settings.

**Reviewer Scores:**

Based on the discussion and rebuttal, most reviewers would likely maintain their original scores:

Reviewers initially below threshold remain unconvinced due to concerns about experimental validity and conceptual framing.

Reviewers initially above threshold acknowledge improvements but still view the contribution as borderline, with unresolved weaknesses.

---

### Decision · Program_Chairs · 2026-01-26

Reject